# Offline Fennel: A High-Performance and Computationally Efficient Biogeochemical Model within the Regional Ocean Modeling System (ROMS)

Júlia Crespin[1], Jordi Solé[2,3], Miquel Canals[4,5,6]

[1]GRC Geociències Marines, Departament de Dinàmica de la Terra i de l'Oceà, Facultat de Ciències de la Terra, Universitat de Barcelona, 08028 Barcelona, Spain.
[2]Institute of Marine Sciences (ICM), Physical and Technological Oceanography Department, Spanish National Research Council (CSIC), 08003 Barcelona, Spain.
[3]Centre for Ecological Research and Forestry Applications (CREAF), 08193 Cerdanyola del Vallès, Spain.
[4]Càtedra d'Economia Blava Sostenible, Universitat de Barcelona, 08028 Barcelona, Spain.
[5]Reial Acadèmia de Ciències i Arts de Barcelona (RACAB), 08001 Barcelona, Spain.
[6]Institut d'Estudis Catalans (IEC), Secció de Ciències i Tecnologia, 08001 Barcelona, Spain.

*Correspondence to*: Júlia Crespin (jcrespin@ub.edu)

**Abstract.** Ocean biogeochemical models are essential for advancing our understanding of oceanographic processes. Here, we present the Offline Fennel model, a biogeochemical model that relies on previously computed physical fields, within the Regional Ocean Modeling System (ROMS). We evaluated the model performance against a fully coupled physical-biogeochemical online application in the Northern Gulf of Mexico, a region with an intense biogeochemical activity including rather frequent hypoxia events. By leveraging physical hydrodynamic outputs, we ran the Offline Fennel model using various time-step multiples from the coupled configuration, significantly enhancing computational efficiency and reducing simulation computational time by up to 87%. The accuracy of the offline model was assessed using three different mixing schemes: the Generic Length Scale (GLS), Large–McWilliams–Doney (LMD, and Mellor and Yamada 2.5 (MY25). The offline model achieved an average skill score of 93%, with minimal impact on performance from the time-step choice. While the GLS configuration yielded the highest accuracy, all three mixing schemes performed well. Although some discrepancies appeared between offline and coupled simulation outputs, these were smaller than those observed when using different mixing schemes within the same model configuration. A significant challenge identified was the simulation of ammonium ($NH_4$), which exhibited the largest discrepancies due to its rapid turnover timescale compared to other tracers. The promising results achieved so far validate the Offline Fennel model's capability and efficiency, thus offering a powerful tool for researchers aiming to conduct extensive biogeochemical simulations without rerunning the hydrodynamic component, thus significantly reducing computational demands.

## 1 Introduction

Ocean biogeochemical models are essential tools for advancing our understanding of oceanographic processes, their impacts on marine ecosystems, and their contributions to climate change projections affecting ecosystem functionality (Aumont et al., 2014; Ramirez-Romero et al., 2020; Fennel et al., 2022). Over recent decades, these models have proven effective in tracking changes in phytoplankton dynamics and nutrient distributions, offering mechanistic insights into ecosystem variability and resilience (Rocha et al., 2019). Additionally, their outputs have had significant applications in fisheries and marine policy, therefore aiding in the development of sustainable management practices (Tedesco et al., 2016; Piroddi et al., 2017).

Despite their proven utility, long-term or high-resolution simulations are often limited by the need of substantial computational resources for coupled configurations, where hydrodynamic and biogeochemical processes are run simultaneously. Offline modeling decouples biogeochemical tracers from the hydrodynamic simulation, thus offering a rather practical alternative (Kim and Khangaonkar, 2012; Larsen et al., 2017). In this approach, outputs from the hydrodynamic model are used as forcing inputs for subsequent biogeochemical simulations, enabling larger computational time-steps (DT) and, therefore, improving efficiency (Thyng et al., 2021). This methodology leverages the slower temporal scales of biogeochemical processes relative to physical dynamics while maintaining simulation quality (Larsen et al., 2017).

An additional challenge in biogeochemical modeling lies in the time-intensive task of fine-tuning biological parameters (Mattern et al., 2017; Pasquier et al., 2023). The offline methodology facilitates more simulations and tests than would typically be possible with fully coupled simulations, as the hydrodynamic component does not need to be rerun each time. This capability is particularly valuable for addressing challenges posed by climate change, allowing for more extensive and exploratory simulations for changing environmental conditions.

Previous studies have explored offline modeling in the Regional Ocean Modelling System (ROMS) framework (Große et al., 2019; 2020), demonstrating its potential for regional and ecosystem-specific applications. In this study, we present the development, implementation, and evaluation of an offline biogeochemical model — hereinafter referred to as "Offline Fennel" (Fennel et al., 2006; Fennel et al., 2008) — within ROMS (Shchepetkin and McWilliams, 2005; Warner et al., 2010).

Building upon the ROMS passive tracer offline code from Thyng et al. (2021), we adapted the model to support biogeochemical processes, incorporating new tracers and parameters critical to ecosystem modeling based on the original Fennel model (Fennel et al. 2006, 2008, 2013; Laurent et al., 2012; Yu et al., 2015a-b) (see Appendix A).

The Offline Fennel model was implemented in the Northern Gulf of Mexico (NGoM), a region known for intense biogeochemical activity and frequent hypoxia events (Rabalais et al., 2002). The NGoM area has been extensively studied

using coupled hydrodynamic-biogeochemical implementations (Fennel et al., 2011, 2013; Laurent et al., 2012, 2014, 2017, 2018; Yu et al., 2015a-b; Fennel and Laurent, 2018; Große et al., 2019). Model performance was evaluated by comparing Offline Fennel outputs against fully coupled simulations, with a focus on assessing the impact of different mixing schemes and time-step (DT) configurations on accuracy and computational efficiency.

This work presents an efficient and accurate tool for biogeochemical modeling, offering enhanced computational efficiency while maintaining simulation fidelity. It provides a valuable resource for conducting extensive simulations and advancing our understanding of complex marine processes.

## 2 Experimental setup

### 2.1 Model overview

ROMS is a hydrostatic, free-surface, terrain-following ocean model that utilizes well-established physical and numerical algorithms (Shchepetkin, 2003). It has been widely applied to simulate various regions of the world ocean (e.g., www.myroms.org/papers). In particular, ROMS coupled configurations have been extensively implemented in the NGoM (Fennel et al., 2011, 2013; Laurent et al., 2012, 2014, 2017, 2018; Yu et al., 2015a-b; Fennel and Laurent, 2018; Große et al., 2019).

In our study we employed a coupled physical-biogeochemical configuration of ROMS (version 904) integrated into the Coupled Ocean–Atmosphere–Wave–Sediment Transport (COAWST) framework (Shchepetkin and McWilliams, 2005; Warner et al., 2010), alongside an offline model implemented within the same version (Thyng et al, 2021). Initially designed for passive tracer applications, the offline model was integrated into ROMS version 904 by Thyng et al. (2021). However, modifications were necessary to adapt it for compatibility with the Fennel biogeochemical model, subsequently enabling accurate representations of biogeochemical processes (**Appendix A**).

A key modification qualified the model to automatically read and process active tracers, namely temperature and salinity, which constitute a crucial feature for biogeochemical simulations. Additionally, improvements were made to fix the handling of climatology files for the bottom-depth layer to ensure more accurate simulations. In the previous model version, a time shift occurred when processing climatology fields, leading to a bias that propagated from the bottom toward the surface, affecting tracer concentrations. The offline model was modified to prevent the simulation from accessing subsequent time step values for sea surface height and 3D momentum climatologies, thereby eliminating this unintended artifact. Further changes included the incorporation of biological tracers such as phosphate ($PO_4$), river carbon detritus, and river nitrogen detritus. The dissolved oxygen ($O_2$) computation method based on Wanninkhof (2014) was also integrated (cf. **Appendix A**).

The version of the biogeochemical model used in this study builds on the ROMS biogeochemical component developed by Fennel et al. (2006, 2008) and later expanded to account for phosphate (Laurent et al., 2012), oxygen (Fennel et al., 2013), and non-sinking river detritus (Yu et al., 2015b). The current biogeochemical model Fennel includes 15 state variables: chlorophyll (CHL), phytoplankton, zooplankton, nitrate ($NO_3$), ammonium ($NH_4$), $PO_4$, $O_2$, dissolved inorganic carbon (DIC), total alkalinity (TA), and three pools of detrital organic matter (small, large, and river-derived, each split into nitrogen and carbon

pools). For further details on the biogeochemical model and parameter values, see Laurent et al. (2017).

## 2.2 Online model setup

The online (or coupled) hydrodynamic-biogeochemical model was configured for the NGoM (28–30.5ºN, 94.5–88ºW) and included 20 terrain-following vertical layers (**Fig. 1**). The horizontal resolution varied from approximately 20 km in the southwestern corner to 1 km close to the Mississippi River delta. The simulation was run for one year (2017-Nov to 2018-

105 Nov) with a baroclinic timestep of 60 s and a barotropic timestep of 15 s.

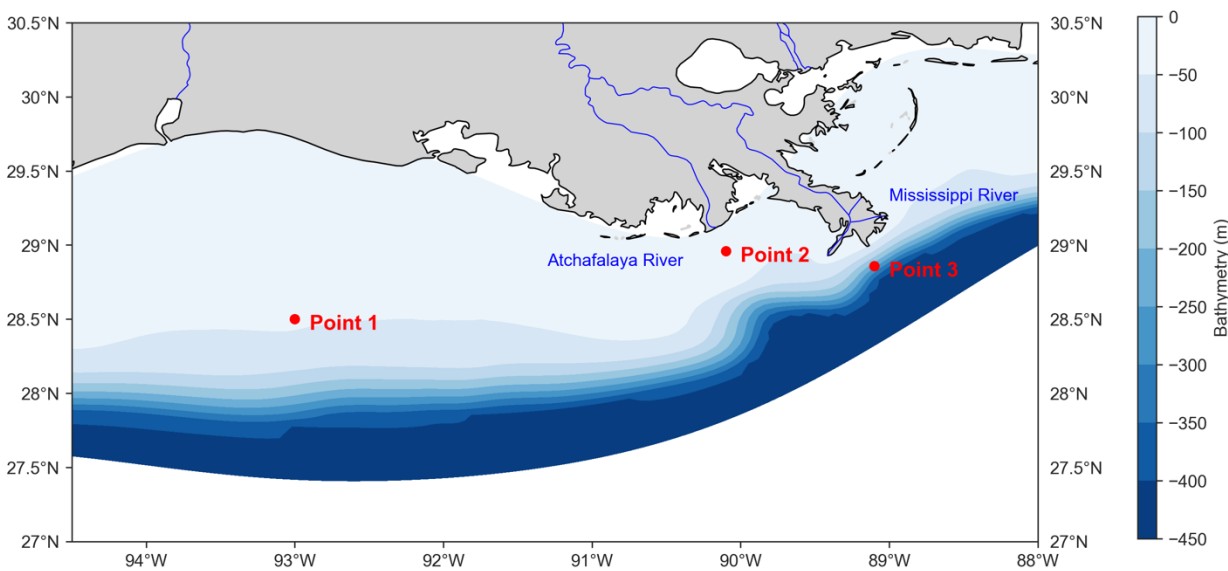

**Figure 1: Model domain in the Northern Gulf of Mexico.** The color scale represents the model's bathymetry (in meters) using a blue color gradient. Red dots indicate the locations selected for vertical profile analyses (from left to right: Point 1, Point 2, and Point 3). The

110 selection criteria for these three points are presented in Section 2.4 of the main text. The Mississippi and Atchafalaya rivers, which are also incorporated into the model, are shown in blue.

Atmospheric forcing included surface heat and freshwater flux climatologies (da Silva et al., 1994a, b), together with 3-hour winds from the National Centers for Environmental Prediction (NCEP) North American Regional Reanalysis data collection

(Mesinger et al., 2006). The U.S. Army Corps of Engineers at Tarbert Landing and Simmesport estimates for freshwater

transports were used to prescribe daily freshwater fluxes from the Mississippi and Atchafalaya rivers, respectively. Additional information about the hydrodynamic model's setup and validation can be found in Hetland and DiMarco (2008, 2012), Marta-Almeida et al. (2013), and Fennel et al. (2016).

To evaluate the offline model performance, online simulations were conducted using three different vertical mixing schemes from ROMS: (i) the Large–McWilliams–Doney (LMD) mixing scheme, also known as the K-Profile Parameterization (Large et al., 1994); (ii) the Mellor and Yamada 2.5 (MY25) scheme (Mellor and Yamada, 1982), which features the "Level 2.5" closure with modifications by Galperin et al. (1988), as detailed in Allen et al. (1995); and (iii) the Generic Length Scale (GLS) mixing scheme, developed by Umlauf and Burchard (2003), which is a versatile two-equation turbulence closure scheme that 125 can be adjusted to replicate several traditional schemes, including MY25. The GLS scheme was integrated into ROMS by Warner et al. (2005). The GLS scheme in our simulations corresponds to the k–ε configuration (Warner et al., 2005), defined by the exponent values 'GLS_P' = 3.0, 'GLS_M' = 1.5, and 'GLS_N'= -1.0.

Harmonic horizontal mixing of velocities and tracers was applied along geopotential surfaces, together with the 130 'TS_MPDATA' advection scheme to minimize numerical diffusion (Thyng et al., 2021). Grid-dependent diffusivity and viscosity were also enabled via 'DIFF_GRID' and 'VISC_GRID'. All configuration files, including exact parameters for each simulation, are available for full reproducibility in a Zenodo repository (Crespin, 2025b).

**2.3 Biogeochemical configuration**

For the biogeochemical implementation, we used the same configuration for both online and offline simulations to ensure 135 comparability. The state variables incorporated all Fennel model tracers except those involved in carbon pools (e.g., total alkalinity).

The initial and boundary conditions for $NO_3$, $PO_4$, and $O_2$ were derived from the National Oceanographic Data Center (NODC) World Ocean Atlas, while all other variables were assigned initial low positive values.

Nutrient sources for rivers were activated using monthly estimates of nutrient fluxes from the U.S. Geological Survey, which provided the basis for river nutrient and organic matter loads (Aulenbach et al., 2007).

**2.4 Offline experiments**

The offline simulations employed the same domain, vertical layers, and horizontal resolutions as the online simulations, 145 ensuring consistency between all configurations. The key difference lies in the physical forcing: while online simulations compute hydrodynamics and biogeochemical processes simultaneously, the offline model derives all physical forcing conditions from the online outputs.

The physical surface forcing conditions for the offline biogeochemical model included solar shortwave radiation flux, surface net heat flux, surface u-momentum stress, and surface v-momentum stress, which were derived from the corresponding online simulation outputs. The climatology forcing incorporated variables such as free surface elevation (zeta), vertically integrated u-momentum component (ubar), vertically integrated v-momentum component (vbar), u-velocity, v-velocity, omega, temperature, and salinity, also retrieved from the historical files of the online simulation outputs. **Appendix B** provides detailed guidance on configuring Offline Fennel simulations.

Following the recommendation of Thyng et al. (2021), a 3-hourly hydrodynamic input frequency was selected to run the offline simulation for both the physical and climatology forcing files. In their Gulf of Mexico simulations, they estimated an advection timescale of approximately 5.6 hours, based on a characteristic velocity of 0.5 m/s and length scale of ~10 km. Our choice of a 3-hour interval thus falls well below this threshold, providing at least one output every ~0.5 advection timescales (Thyng et al., 2021). This frequency is therefore adequate to resolve key physical changes and ensure accurate offline interpolation of biogeochemical tracers.

A series of online (coupled) and offline (uncoupled) simulations were conducted using three different vertical mixing schemes: LMD, MY25, and GLS (cf. **Section 2.2**). All simulations were stored as instantaneous snapshots (saved in ".his" files from ROMS) and as time-averaged data (".avg" files from ROMS). The output frequency for both instantaneous and averaged files was 3 h.

For the GLS and MY25 simulations, the offline model was forced by additional vertical mixing parameters – namely vertical salinity diffusion (AKs), temperature vertical diffusion coefficient (AKt), and vertical viscosity coefficient (AKv), –, all of which influence sub-grid-scale vertical mixing. These fields are obtained from the online parent run via the 'AKXCLIMATOLOGY' CPP flag, which ingests the 3-h climatologies of AKs, AKt, and AKv. In addition, the 'MIXCLIMATOLOGY' flag was used to import the generic length scale (GLS) and turbulent kinetic energy (TKE) coefficients from the online simulation.

In contrast, LMD simulations omit these climatology flags, so the offline model recomputes its own AKs, AKt, and AKv internally by using the same turbulence-closure parameters defined in the online configuration. This ensures that vertical diffusivity remains active under all schemes while enabling a direct test of sensitivity to externally prescribed versus internally computed mixing fields. This treatment also mirrors the approach of Thyng et al. (2021) for the GLS and MY25 cases; for full implementation details of these flags, refer to Thyng et al. (2021).

Offline simulations were run with varying multiples of the online DT (x1, x3, x5, x10, and x15) to improve computational efficiency, until the results became unstable. Given that the baroclinic DT of the online simulation was 60 seconds, these corresponded to offline baroclinic DTs of 60 s, 180 s, 300 s, 600 s and 900 s. A DT 15 times longer than the online time-step led to unstable writing of solutions. As such, while this case was initially tested, it was excluded from analysis figures and tables to avoid misleading interpretations.

All skill assessments in this study compare the outputs of the offline simulation to those of the online (coupled) simulation that provided the physical forcing for the offline run. As such, the study reflects the uncoupled simulation's accuracy with respect to the coupled simulation. Therefore, this is an assessment on the offline model's ability to reproduce the coupled simulation results, instead of an assessment on how well the NGoM biogeochemistry is simulated by the models.

To assess the vertical accuracy in the offline simulation results, we selected three representative points in the NGoM based on their distinct environmental and geographic characteristics. The first point, located in the western part of the study area and far from the coast (28.50ºN, 93.00ºW, Point 1), was chosen to represent offshore conditions with minimal direct riverine influence. The second point near the mouth of the Atchafalaya River (28.96ºN, 90.10ºW, Point 2) captures the influence of a significant freshwater and nutrient source, providing insights into river-plume dynamics. The third point is located further east, off the Mississippi River mouth (28.86ºN, 89.10ºW, Point 3), and was selected to represent a highly dynamic coastal environment influenced by one of the world's largest river systems (**Fig. 1**). These three points collectively offer a comprehensive view of physical and biogeochemical gradients in the region, capturing offshore, plume-affected, and coastal conditions.

All the time series presented in the Results section have been upscaled to daily data to enhance the visibility of changes and variability. This adjustment allows for a clearer observation of trends and fluctuations over time.

Considering that we are comparing 3-D time-varying model results for multiple variables, the potential number of plots grows rapidly, thus exceeding what could be reasonably presented in a single paper. Therefore, we focus here on three principal nutrients ($NO_3$, $NH_4$, $PO_4$), CHL, and $O_2$ to assess the model performance, even though all variables described in **Section 2.3** were used in the model implementation.

### 2.5 Skill metrics

We evaluate the Offline Fennel model using two complementary, volume-weighted diagnostics: a skill score (SS) and the root-mean-square error (RMSE). Both metrics account for the true physical volume ($V_{i,j,k}$) of each grid cell, thereby avoiding biases due to varying horizontal areas or layer thicknesses.

Each cell volume ($V_{i,j,k}$) is computed as the product of the horizontal ROMS grid spacings ($\Delta x_i$, $\Delta y_j$) and the vertical thickness ($\Delta z_{i,j,k}$), which is calculated from the difference in model layer depths ($z_w$) at each horizontal location ($i,j$).

SSs are a widely used metric for evaluating model performances (Bogden et al., 1996; Hetland, 2006). To assess the performance of our Offline Fennel model, we applied the following equation (**Eq. 1**), adapted from Thyng et al. (2021):

$$SS = 1 - \sqrt{\frac{\sum_{i,j,k} V_{i,j,k} \times \left(C(t) - C_{ref}(t)\right)^2}{\sum_{i,j,k} V_{i,j,k} \times C_{ref}(t)^2}} \quad \text{(Eq. 1)}$$

where $C(t)$ and $C_{ref}(t)$ are the concentrations of a tracer of a tracer at time $t$ in the compared and reference simulations, respectively – typically representing offline (uncoupled) and online (coupled) configurations. The sums are performed over all spatial and vertical dimensions, and results are volume-weighted. This yields a time series of SS values that tracks the temporal evolution of model performance. A time-mean SS can be computed by averaging over the simulation period, providing a single, scalar measure of overall model accuracy.

To complement the SS analysis, RMSE was equally employed as a metric to assess the accuracy of offline simulations compared to online results (**Eq. 2**). RMSE provides insight into the magnitude of errors by measuring the square root of the average squared differences between offline and online simulation results.

$$RMSE = \sqrt{\frac{\sum_{i,j,k} V_{i,j,k} \times \left(C - C_{ref}\right)^2}{\sum_{i,j,k} V_{i,j,k}}}$$

(Eq. 2)

where $C$ and $C_{ref}$ are the time-averaged concentrations of a tracer on the 3D grid for the offline and online simulations, respectively. Because each grid cell's contribution is proportional to its volume, this RMSE reflects the true three-dimensional error structure.

Together, these volume-weighted SS and RMSE metrics provide a robust evaluation of model performance, capturing both relative and absolute discrepancies while avoiding biases caused by unequal grid cell sizes.

## 3 Results

This section presents the evaluation of the Offline Fennel model against fully coupled online configurations. The first subsection provides an assessment of the model's accuracy using SS and RMSE, highlighting the agreement between simulation outputs. Then, key biogeochemical outputs are examined, including nutrients, CHL, and $O_2$ levels, to explore spatial and temporal variations between the simulation methods. Finally, the computational efficiency of the offline approach is analysed, demonstrating its potential for reducing simulation runtime without compromising the quality of the results.

In this study, we define the 'surface layer' as the uppermost of the model's 20 vertical layers (cf. **Section 2.2**), crucial for atmosphere-ocean interactions, primary production, and gas exchange. The 'bottom layer,' in direct contact with the sea floor, is key for assessing the model's response to bathymetry. Since these layers are most susceptible to error propagation, their evaluation is essential for validating model performance.

### 3.1 Model performance evaluation

Here we evaluate the accuracy of offline simulations using the three mixing schemes (GLS, LMD, and MY25) outlined in **Section 2.2**, together with variable offline DTs, which are multiples of the online DTs (x1, x3, x5, and x10). It should be noted that the model configuration is identical in all cases, with the mixing scheme varying in both the online and offline simulations, and the DT varying only in the offline simulations.

To assess model performance, we first present a Taylor diagram (**Fig. 2**) (Taylor, 2001) that illustrates the volume-weighted and normalized statistics averaged across all biogeochemical variables. This diagram highlights a strong agreement between the offline simulations and the online parent model. The offline configurations show higher standard deviation values compared to the coupled reference (0.26 across all vertical mixing schemes), exhibiting relative standard deviations slightly exceeding 1.

GLS and LMD demonstrate remarkably similar performance, characterized by standard deviations ranging from 1.107 to 1.123, low centered root mean square error (RMSE) values around 0.25, and high correlation coefficients (r > 0.98). The MY25 scheme shows slightly higher RMSE values (up to 0.29) and marginally lower correlation coefficients, but these differences are minimal and do not significantly detract from its overall performance. Furthermore, differences between DTs are negligible across all cases (**Fig. 2**).

**Table 1** summarizes the mean SSs computed using **Eq. 1** for key biogeochemical tracers. Across all mixing schemes, the simulations demonstrate high accuracy, with minimal differences between configurations. GLS slightly outperforms the others in some tracers, with scores above 95% for $NO_3$, $PO_4$, and $O_2$, and a mean SS of 92.92% across all tracers. CHL scores hover

around 91%, highlighting the scheme's ability to capture primary production dynamics effectively. However, $NH_4$ exhibits lower SSs, ranging from 83.21% to 83.53%.

The LMD scheme similarly produces robust results, particularly for $O_2$, $PO_4$, and $NO_3$ with SSs of 98.68%, 96.00% and 95.44%, respectively. Its mean SS is slightly lower (92.79%), reflecting solid tracer performance overall, although $NH_4$ again shows the weakest accuracy, with SSs between 82.53% and 82.94%.

Results for the MY25 scheme align closely with those of GLS and LMD, yielding SSs of 96.12% for $NO_3$ and 96.86% for
$PO_4$. The mean SS for MY25 is 92.86%, underscoring its comparable performance. While $NH_4$ again exhibits lower SSs, the values still represent good model performance, as they remain above 80%. This suggests that although $NH_4$ is more challenging compared to other tracers, the model still provides a reliable approximation of its concentrations.

Overall, the analysis reveals that all three mixing schemes (GLS, MY25, and LMD) yield comparable and robust results for
key biogeochemical tracers, with GLS performing slightly better in some metrics, followed closely by MY25 and LMD.

Finally, variations in the offline DTs had minimal impact on the model's accuracy. This suggests that the choice of DT, at least within the tested range, does not significantly influence simulation results, underscoring the robustness of the Offline Fennel model.

The x5 DT configuration consistently yields the highest accuracy across all mixing schemes, as illustrated in **Figure 3**, which shows the SS evolution over time for key biogeochemical tracers. As expected, $NO_3$, $PO_4$, and $O_2$ maintain the highest accuracy throughout the entire simulation period, with SSs ranging from 90% to 99%.

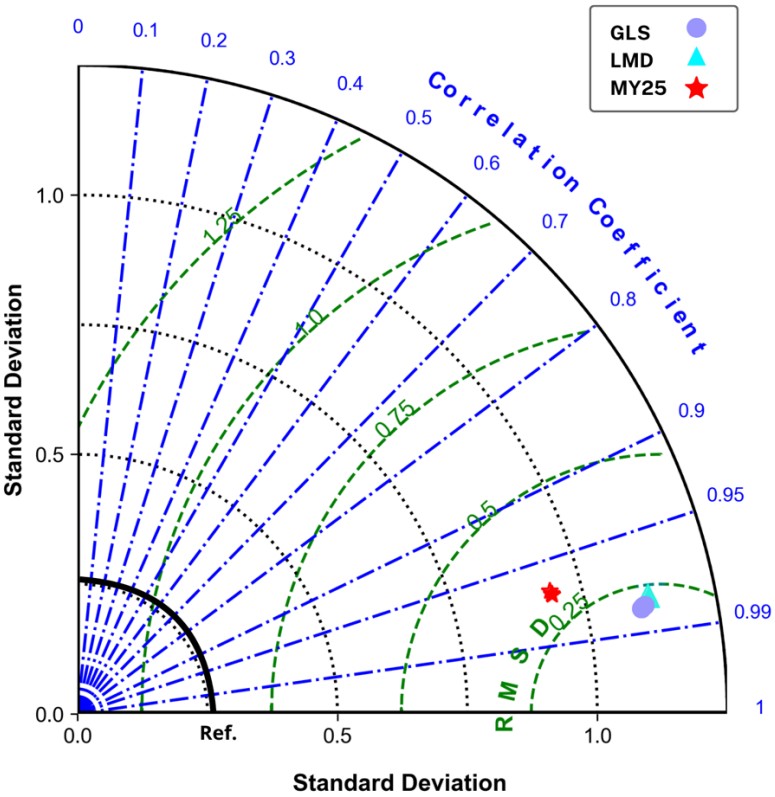

**Figure 2. Taylor diagram illustrating the comparison of standard deviations, centered root mean square differences (RMSD), and correlation coefficients across various offline experiments relative to online reference data.** Each point represents the performance metrics of different models: GLS metrics are indicated by purple circles, LMD by cyan triangles, and MY25 by red stars, plotted across different time scales (DTs: x1, x3, x5, and x10). The thick black semi-circle denotes the reference (Ref.) standard deviation from the online

experiments.

In contrast, NH₄ and CHL exhibit lower accuracy compared to other tracers. Notably, NH₄ accuracy declines from 90% in early spring to 70% during the warmer months between April and October, with the LMD scheme showing the sharpest drop, reaching as low as 61% in September. Similarly, CHL accuracy dips from 95% to approximately 85% during April and May.

For NH₄, the GLS scheme tends to perform slightly better, with higher SS values across most DT configurations (**Figs. S1, S2, S3**). For other tracers, differences among the schemes are minimal, and no consistent ranking is evident.

To better illustrate spatial differences, we calculated the RMSE across depth for key biogeochemical variables (**Eq. 2**) in the NGoM, considering all three mixing schemes for the offline x5 DT simulation to ensure consistency with other figures (**Fig.**

**4**).

**Table 1: Time-averaged skill scores (SSs) for key biogeochemical (BGC) tracers across different mixing schemes: Generic Length Scale (GLS), Large–McWilliams–Doney (LMD), and Mellor and Yamada 2.5 (MY25).** The SSs are expressed as percentages [%] and reflect the model's performance in simulating the following biogeochemical (BGC) tracers: nitrate ($NO_3$), ammonium ($NH_4$), phosphate ($PO_4$), chlorophyll (CHL), and oxygen ($O_2$). The last row for each mixing scheme displays the mean SSs for that scheme, providing an overall assessment of model performance across all tracers. The columns labeled x1, x3, x5, and x10 correspond to the time-steps (DT) used in the offline simulations compared to the online simulations.

| Mixing scheme | BGC tracer | x1 [%] | x3 [%] | x5 [%] | x10 [%] |
|---|---|---|---|---|---|
| GLS | $O_2$ | 98.70 | 98.72 | 98.72 | 98.69 |
| | $PO_4$ | 96.08 | 96.10 | 96.11 | 96.08 |
| | $NO_3$ | 95.62 | 95.65 | 95.67 | 95.67 |
| | CHL | 90.88 | 90.99 | 90.94 | 90.37 |
| | $NH_4$ | 83.21 | 83.33 | 83.42 | 83.53 |
| | Mean | 92.90 | 92.96 | 92.97 | 92.87 |
| LMD | $O_2$ | 98.68 | 98.69 | 98.68 | 98.62 |
| | $PO_4$ | 96.00 | 96.01 | 95.99 | 95.94 |
| | $NO_3$ | 95.44 | 95.45 | 95.44 | 95.37 |
| | CHL | 91.23 | 91.36 | 91.27 | 90.49 |
| | $NH_4$ | 82.53 | 82.73 | 82.85 | 82.94 |
| | Mean | 92.78 | 92.85 | 92.85 | 92.67 |
| MY25 | $O_2$ | 98.71 | 98.72 | 98.71 | 98.66 |
| | $PO_4$ | 96.20 | 96.21 | 96.21 | 96.16 |
| | $NO_3$ | 95.50 | 95.51 | 95.51 | 95.47 |
| | CHL | 91.39 | 91.49 | 91.39 | 90.57 |
| | $NH_4$ | 82.43 | 82.64 | 82.78 | 82.92 |
| | Mean | 92.85 | 92.92 | 92.92 | 92.76 |

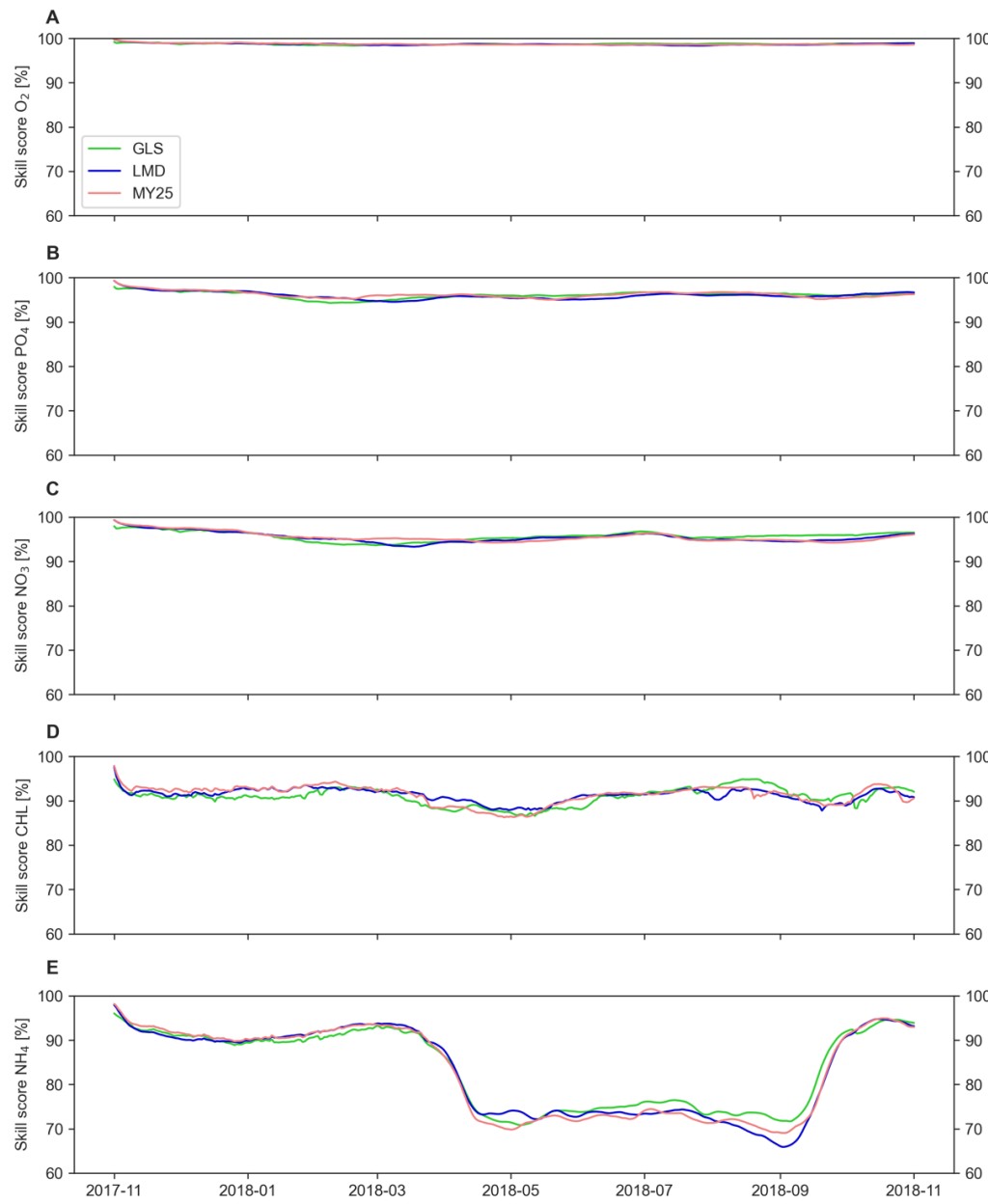

**Figure 3: Time series of volume-weighted skill scores (SSs) [%] for key biogeochemical tracers during the x5 time-step simulation.** The score is computed between offline and online runs using three mixing schemes: Generic Length Scale (GLS), Mellor–Yamada 2.5 (MY25), and Large–McWilliams–Doney (LMD). Panels display (A) dissolved oxygen ($O_2$), (B) phosphate ($PO_4$), (C) nitrate ($NO_3$), (D) chlorophyll (CHL), and (E) ammonium ($NH_4$). The green, blue, and coral lines represent GLS, LMD, and MY25 simulations, respectively.

In relation to typical tracer concentrations in the study area, $O_2$ systematically shows the lowest errors across the domain, with only small and localized increases near the coast. Similarly, CHL errors are generally low but increase near coastal regions, particularly close to the Atchafalaya and Mississippi river mouths (cf. **Fig. 1**). $NO_3$ and $PO_4$ display slightly higher errors near the coast and offshore, with some current-related discrepancies to the south of the domain.

Finally, $NH_4$ presents the highest error levels when considering typical concentrations in the region, which range from 0 to 5 mmol·m$^{-3}$ at the surface and can reach up to 20 mmol·m$^{-3}$ at depth. Among the mixing schemes, GLS shows somewhat lower errors for $NH_4$ specially near the coast (**Fig. 4A**), but overall, all three schemes display similar behavior, particularly in offshore regions where errors approach zero.

Crossing all the time series from each of the simulations and applying the SS calculation for each pair, allows generating the heatmap shown in **Figure 5**. The figure illustrates that simulations using the same mixing scheme exhibit the highest similarity, with SSs ranging between 95% and 100%. In contrast, comparisons across different mixing schemes show decreased SSs, dropping to 92% in some cases between GLS and MY25 simulations. Notably, comparisons between GLS and MY25, and LMD and MY25 show only minor differences.

These results indicate that the choice of mixing scheme has a more substantial impact on simulation accuracy than the offline biogeochemical model configuration itself and its chosen DT.

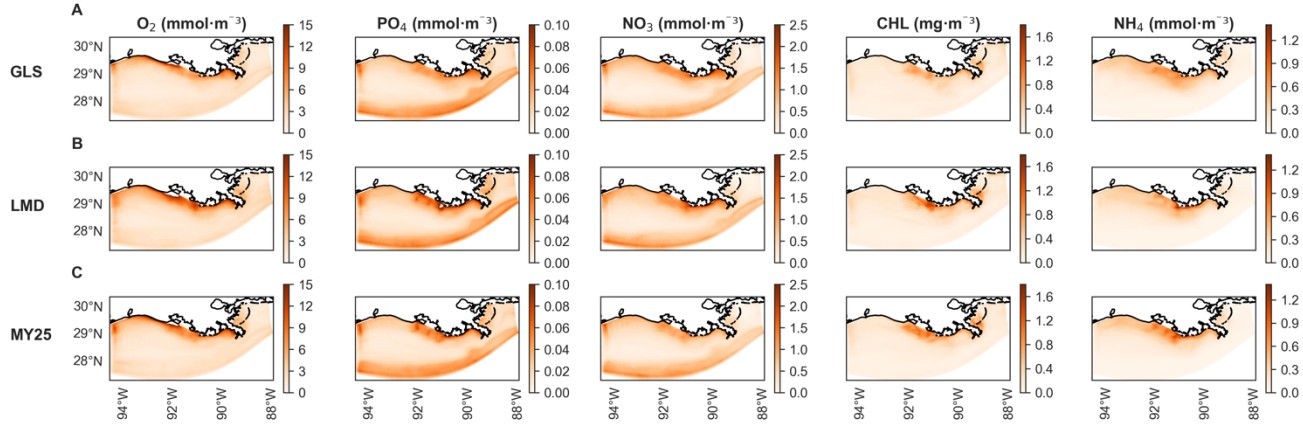

**Figure 4: Mean root mean square error (RMSE) across depth (volume-weighted) for key biogeochemical variables: dissolved oxygen ($O_2$), phosphate ($PO_4$), nitrate ($NO_3$), chlorophyll (CHL), and ammonium ($NH_4$).** The error is calculated between offline (x5 time-step) and online simulations using different mixing schemes: (A) Generic Length Scale (GLS), (B) Large–McWilliams–Doney (LMD), and (C) Mellor–Yamada 2.5 (MY25). The color scales on the right indicate the magnitude of the error.

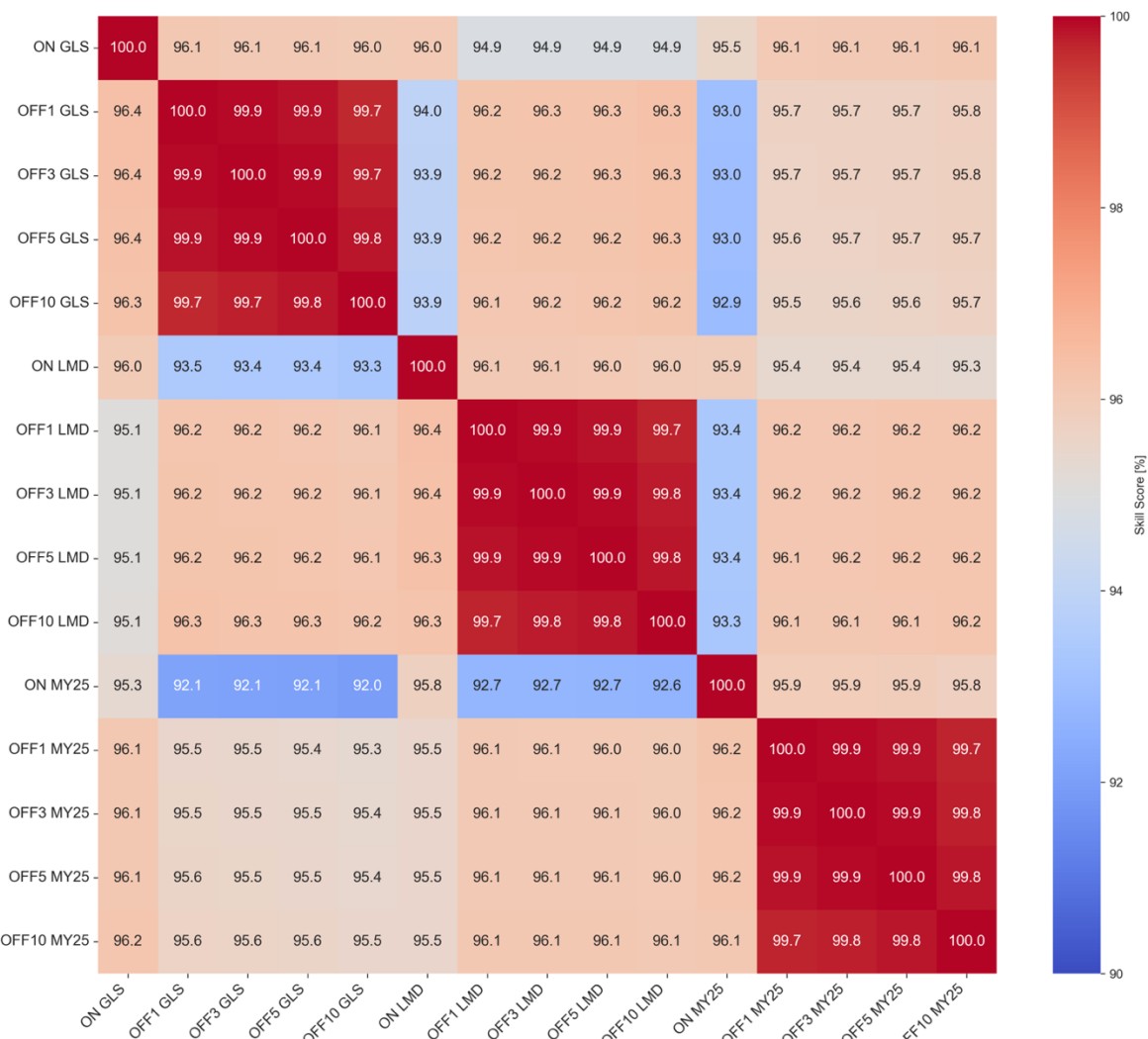

**Figure 5: Heatmap illustrating volume-weighted skill scores (SSs) [%] for all simulation pairs across different mixing schemes and time steps (DT).** The color scale ranges from blue (indicating lower SSs) to red (indicating higher SSs), with warmer colors representing SSs closer to 100%, reflecting greater similarity between simulations. Each label in the heatmap indicates the simulation type, mixing scheme, and DT. For example, "ON GLS" refers to online simulations using the Generic Length Scale (GLS) mixing scheme, while "OFF GLS x5" corresponds to offline simulations using the GLS mixing scheme with a x5 DT. This labeling convention is consistently applied across all mixing schemes: GLS, Large–McWilliams–Doney (LMD), and Mellor–Yamada 2.5 (MY25), and DTs (x1, x3, x5, and x10), facilitating easy comparison of model performance across different configurations. The heatmap highlights consistently higher SSs within simulations that utilize the same mixing scheme, regardless of whether they are online or offline.

### 3.2 Comparison of biogeochemical outputs from offline and online simulations

Here we present the performance of selected offline simulations vs. online simulations. Given that multiple simulations were conducted, we highlight key results that effectively demonstrate the offline model performance across different mixing schemes. For consistency, all plots were generated using 3-hourly 'avg' output files from ROMS, which were then daily averaged. When showcasing a single offline configuration, we selected the GLS mixing scheme with a x5 DT, which was arbitrarily chosen mainly to ensure uniformity across the analysis. The focus is, therefore, on assessing the ability of the offline model to reproduce the online model simulation results.

The main plots presented here focus on the surface layer, which is the most dynamically variable region due to exposure to the physical model forcing (e.g., shortwave solar radiation, net heat flux, u- and v- momentum). This variability increases the likelihood of discrepancies between the online and offline simulations. Secondarily, the complementary figures analyze the bottom layer, which can also exhibit bias due to its interaction with the bathymetry. Additionally, we examine vertical profiles to ensure that the entire water column is accurately represented.

**Figure 6A-D** illustrates the time series of key biogeochemical variables ($O_2$, $PO_4$, $NO_3$, CHL, and $NH_4$) for both the online simulations and the offline simulations at the surface layer with a DT x5. The results show near-perfect alignment between both simulations, with equal or nearly equal time evolution and trends in most cases. Differences appear only for $NH_4$, which is overestimated during the April-October period, especially with the LMD scheme (**Fig. 6B**). Minor differences are also observed in CHL concentrations during the same period (**Fig. 6D**).

A similar behavior is observed when analyzing the time series for the bottom layer (**Fig. S4**), with a very well reproduced temporal evolution of key variables and practically negligible differences. However, $NH_4$ discrepancies become more pronounced (**Fig. S4B**).

**Figure 7** presents the averaged spatial differences in surface layer concentrations. Discrepancies are generally small across all three schemes, with GLS showing slightly smaller differences in some regions (**Fig. 7A**). Near-coastal areas show an underestimation of $O_2$, $PO_4$, and $NO_3$, whereas CHL and $NH_4$ are overestimated in the coastal regions, with accurate representation in the southern region. $PO_4$ shows negligible overestimation (maximum difference of +0.04 mmol·m$^{-3}$) in the southernmost region of the domain. Despite these spatial differences, the error magnitudes remain within acceptable limits and align with typical concentrations of the key biogeochemical tracers in the NGoM (Fennel et al., 2011; Fennel and Laurent, 2018).

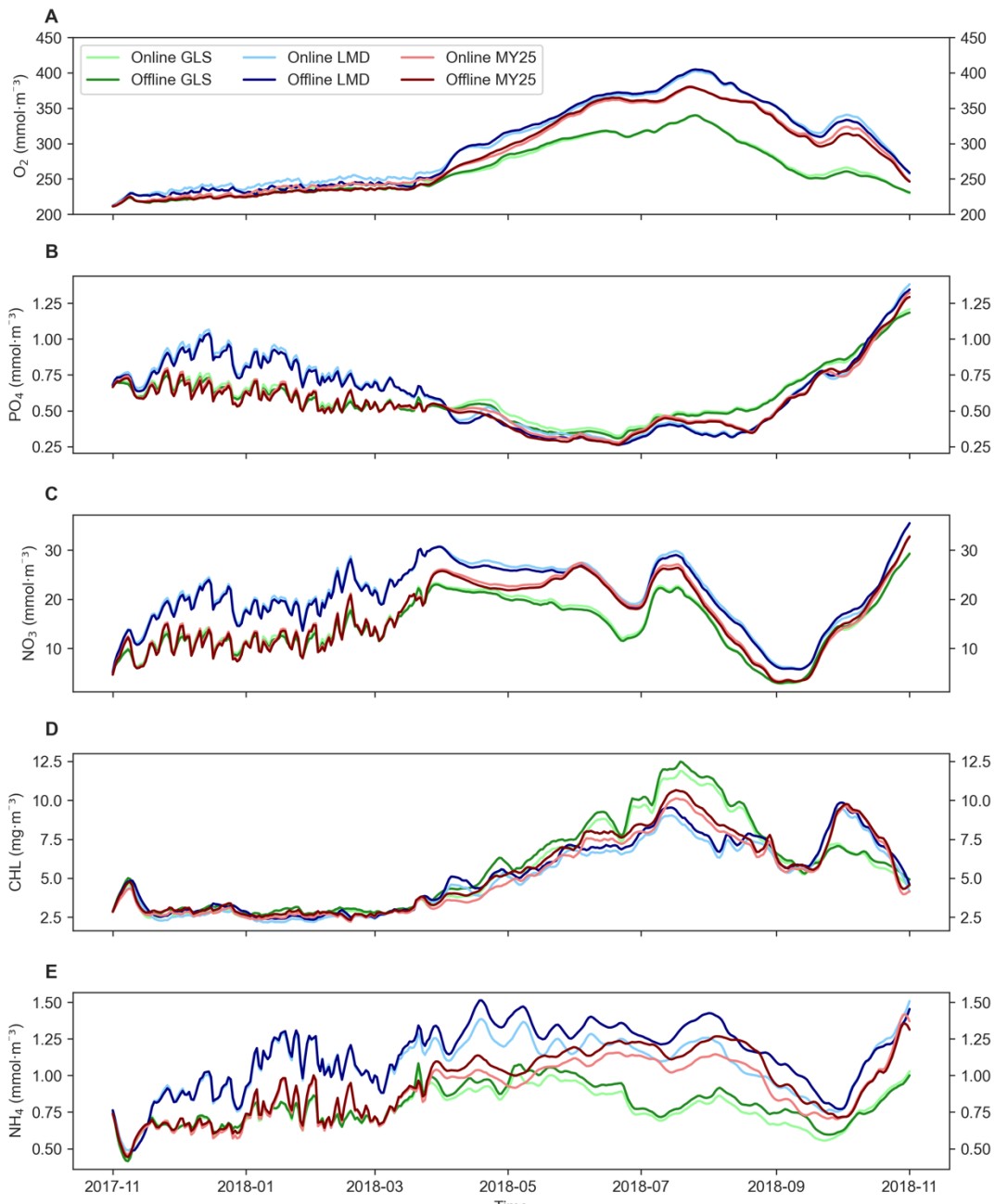

**Figure 6: Area-averaged time series of biogeochemical variables at the surface layer for x5 time-step online and offline simulations using different mixing schemes.** Panels show (A) dissolved oxygen ($O_2$), (B) phosphate ($PO_4$), (C) nitrate ($NO_3$), (D) chlorophyll (CHL), and (E) ammonium ($NH_4$). Light and dark green, light and dark blue, and light and dark red represent online and offline simulations that used Generic Length Scale (GLS), Mellor–Yamada 2.5 (MY25), and Large–McWilliams–Doney (LMD) mixing schemes, respectively.

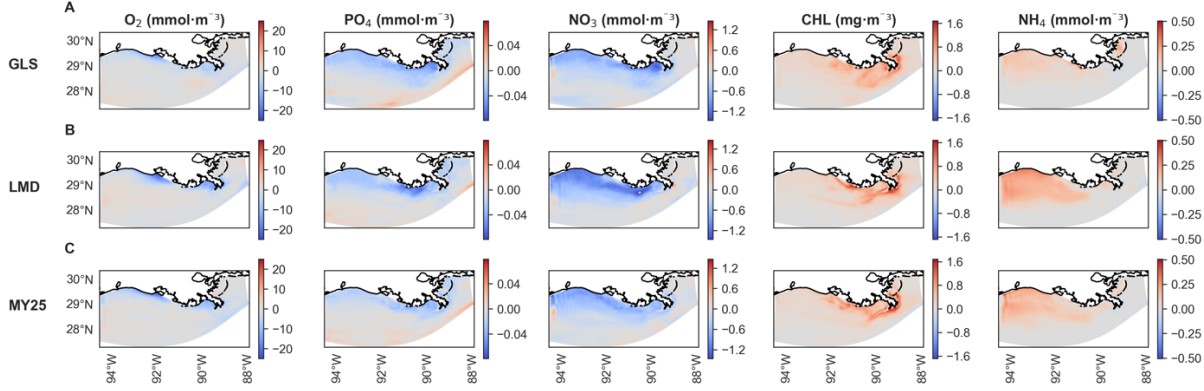

**Figure 7: Time-averaged spatial differences in concentrations of dissolved oxygen (O₂), PO₄ (phosphate), NO₃ (nitrate), CHL (chlorophyll), and NH₄ (ammonium), at the surface layer between offline (x5 time-step) and online simulations using different mixing schemes.** Rows show biogeochemical tracers for (A) Generic Length Scale (GLS) mixing, (B) Large–McWilliams–Doney (LMD) mixing, and (C) Mellor–Yamada 2.5 (MY25) mixing. The 'coolwarm' color scales on the right illustrate the differences.

When examining time-averaged vertical profiles of key biogeochemical variables at various points within the domain (**Fig. 1**), differences between offline simulations with varying DTs are either minimal or negligible (**Fig. 8A-C**).

For O₂, the offline model demonstrates consistent accuracy across all three profiles in points 1, 2, and 3, indicating reliable performance throughout the full domain (**Fig. 8A-C**). Regarding PO₄ and NO₃, the offline model also shows a very good match with online outputs, with only a slight underestimation in Point 1 (offshore) (**Fig. 8A**).

In terms of CHL, the offline model perfectly matches the online model output at points 2 and 3 (near the mouth of the Atchafalaya and the Mississippi rivers, respectively) (**Fig. 8B and C**). However, a slight overestimation is found within the first 15 m at point 1 (offshore), where CHL concentrations are low, so a small bias is expected. This becomes more pronounced as the DT increases (**Fig. 8A**).

For NH₄, it reproduces its vertical behavior almost perfectly at point 3 (**Fig. 8C**), while overestimations are observed at points 1 and 2 (**Fig. 8A-B**), located offshore and near the mouth of the Atchafalaya River, respectively (**Fig. 1**). This overestimation is more pronounced near the Atchafalaya River, which is expected due to rivers exhibiting greater variability. However, despite these overestimations, the overall vertical patterns remain accurate.

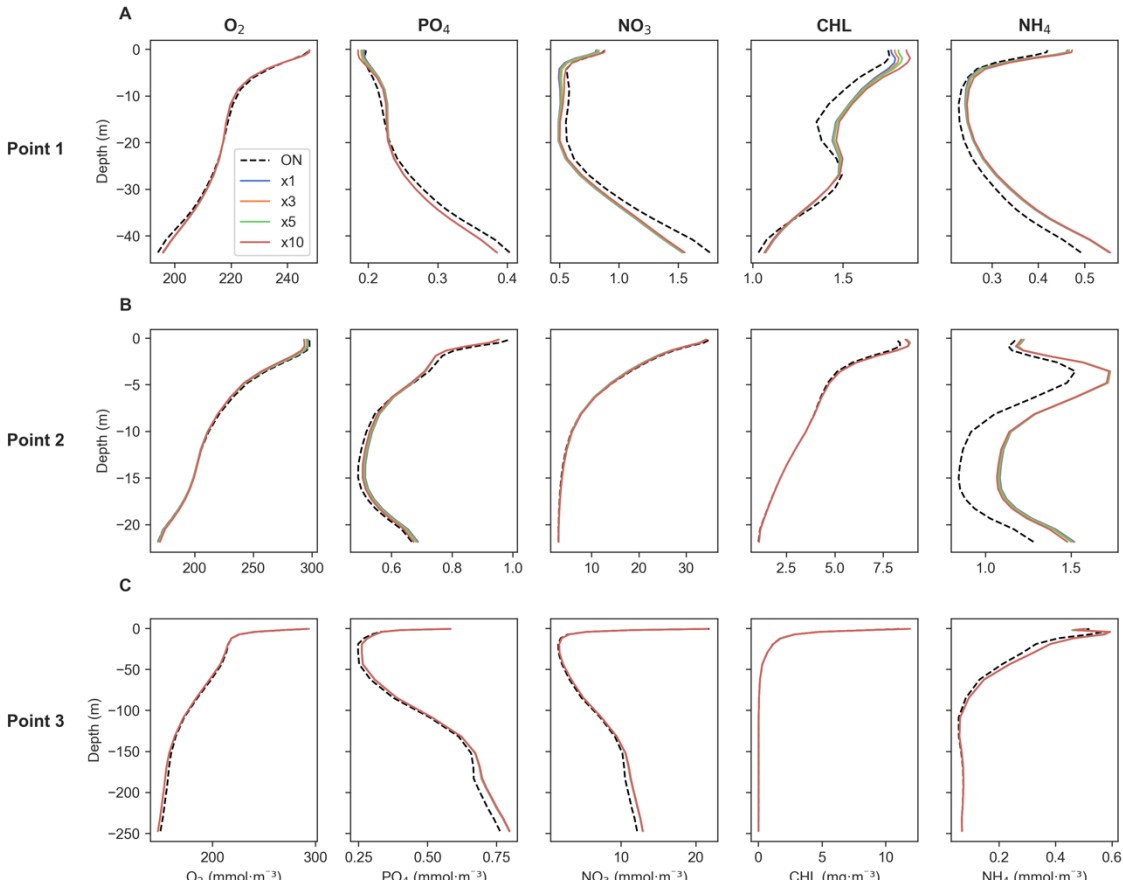

**Figure 8: Time-averaged profiles of key biogeochemical tracers' concentrations for Generic Length Scale (GLS) mixing simulations.** Panels show (A) Point 1, offshore in the western part of the study area; (B) Point 2, near the mouth of the Atchafalaya River; and (C) Point 3, near the mouth of the Mississippi River. Refer to Figure 1 for point locations. The black dashed line represents online (ON) simulations, while the colored lines indicate offline simulations for time steps x1 (blue), x3 (orange), x5 (green), and x10 (red).

Following the examination of biogeochemical tracers, we now turn our attention to surface spatial differences in CHL concentrations. This focus is particularly relevant, as it serves as a critical indicator of primary productivity, which depends on nutrient concentrations such as $NO_3$, $NH_4$, and $PO_4$.

Seasonal CHL concentration maps derived from the GLS mixing simulations illustrate the differences across seasons. Both the online GLS simulation and the corresponding offline GLS simulation with a x5 DT exhibit virtually no discernible differences, displaying identical seasonal spatial patterns (**Figs. 9A, B and C**). The maximum overestimation is of 4 mg·m$^{-3}$ during the summer (June-July-August; JJA) and spring (March-April-May; MAM) seasons (**Fig. 9A**). These are primarily

concentrated near the mouths of rivers that flow into the NGoM, indicating the influence of riverine inputs on CHL concentrations.

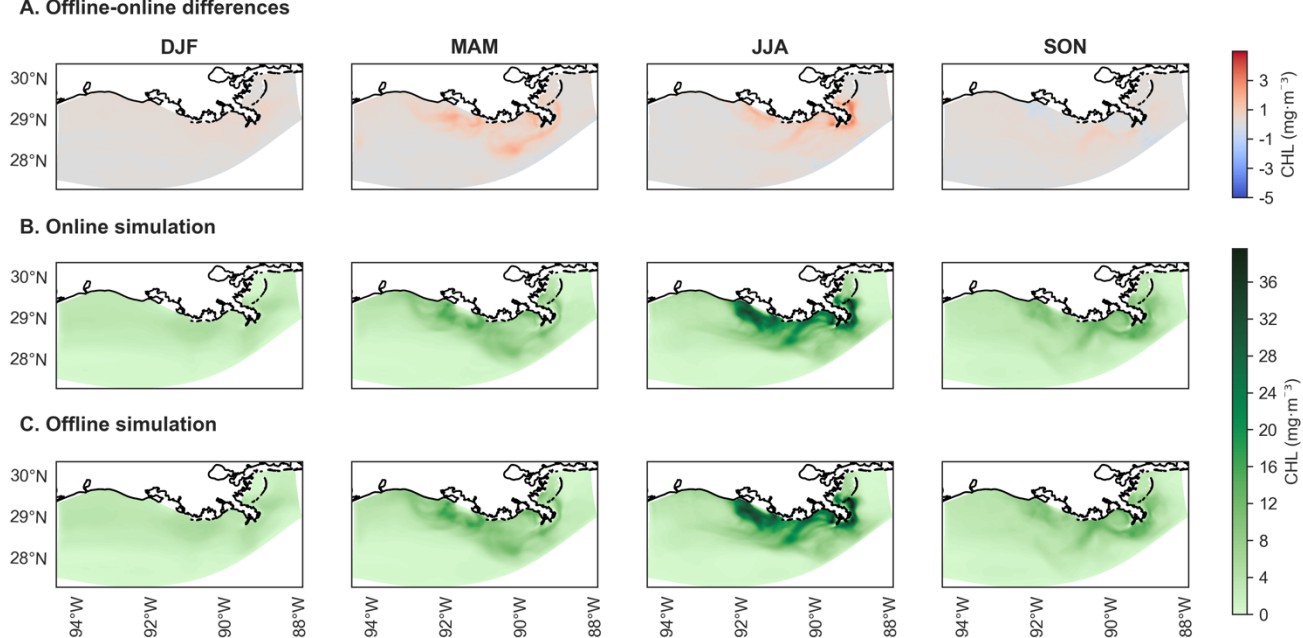

**Figure 9: Seasonal maps of chlorophyll concentrations (CHL) for Generic Length Scale (GLS) mixing simulations using x5 time-step (DT).** (A) Displays the difference in concentrations between offline (x5 DT) and online simulations, as indicated by the coolwarm color scale on the right. (B) Shows chlorophyll concentrations for the online simulation, while (C) presents concentrations for the offline simulation (x5 DT). Panels (B) and (C) share the same color scale from the *cmocean* package ('algae') (Thyng et al., 2016). Seasonal designations are as follows: DJF (December-January-February, winter), MAM (March-April-May, spring), JJA (June-July-August, summer), and SON (September-October-November, fall).

Looking at the other mixing schemes in terms seasonal variations, the winter (December-January-February; DJF) and fall (September-October-November; SON) seasons exhibit minimal variation, but the three mixing schemes produce largely similar patterns (**Fig. S5**). While LMD appears to show slightly more variability in fall, these differences are modest (**Fig. S5**).

In spring, all three schemes show overestimated CHL concentrations, with values reaching up to 2 mg·m⁻³ near the coast. This overestimation peaks during summer, with differences approaching 5 mg·m⁻³. Among the schemes, GLS tends to yield slightly lower deviations in some locations, although all remain within a comparable range.

Differences in CHL concentrations in the bottom layer are also observed, which correspond both to rather small over and underestimations (**Fig. S6**). A consistent overestimation appears in the Atchafalaya Bay region and in the northeastern part of

the domain across all four seasons, with biases reaching up to 1 mg·m⁻³. The northwestern region of the domain displays a combination of underestimations, particularly during spring and fall, with differences of up to 0.5 mg·m⁻³, and slight overestimations in summer.

The offline model also shows strong overall performance across all schemes when examining seasonal CHL differences in vertical profiles (**Fig. 10**). No scheme in particular outperforms the others. For instance, the GLS scheme exhibits a slight overestimation during the fall at Point 1 (**Fig. 10A**) and in winter at Point 2 (**Fig. 10B**). Meanwhile, MY25 and LMD demonstrate comparable deviations throughout the year (**Fig. 10**). Overall, the results indicate that all schemes effectively reproduce vertical CHL patterns, with only minor and spatially variable differences.

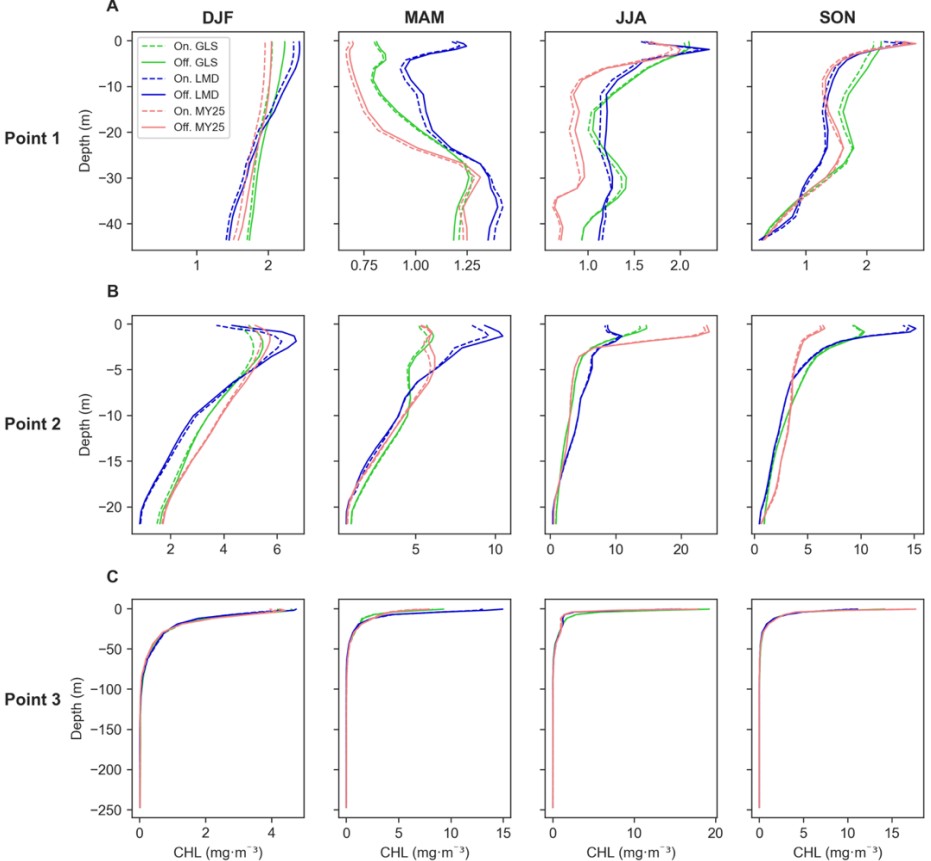

**Figure 10: Seasonal-averaged chlorophyll concentration (CHL) profiles for offline (x5 time step, solid line, Off.) and online simulations (dashed line, On.) across different mixing schemes.** Panels display data for: (A) Point 1, located offshore in the western part of the study area; (B) Point 2, situated near the mouth of the Atchafalaya River; and (C) Point 3, located near the mouth of the Mississippi River. Refer to Figure 1 for specific point locations. The green, blue, and red lines represent the Generic Length Scale (GLS), Large–

470 McWilliams–Doney (LMD), and Mellor-Yamada 2.5 (MY25) mixing schemes, respectively. Seasonal designations are as follows: DJF

(December-January-February, winter), MAM (March-April-May, spring), JJA (June-July-August, summer), and SON (September-October-November, fall).

### 3.3 Computational efficiency analysis

Assessing the computational efficiency of the offline model and the impact of DT increases on the results of the simulation for each mixing scheme is of the utmost importance. Variations in computational time across simulations using the same cluster and node configuration are illustrated in **Figure 11**. The coupled online simulations, which integrate hydrodynamic and biogeochemical components, ranged from 4 hours 36 minutes for the LMD scheme to 5 hours 52 minutes for the MY25 scheme (**Table S1**). Once the offline model was implemented using the same DT than the online configuration, simulation time
decreased by 39% for MY25, 33% for GLS, and 25% for LMD. Further time reductions were achieved by increasing the DT: x3 reduced the simulation time by 65% on average, x5 by 75%, and x10 by 87%.

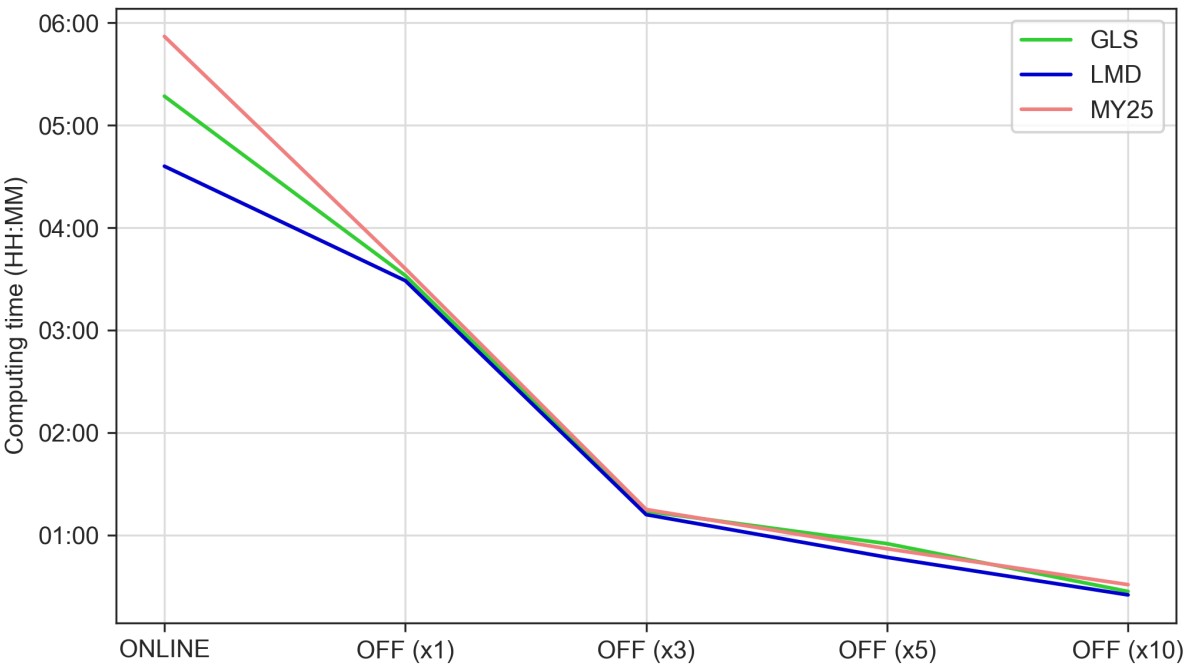

**Figure 11: Computing time (hh:mm) for online and offline simulations across each configuration, considering the three mixing**
**schemes.** The color coding corresponds to the mixing schemes: green for Generic Length Scale (GLS), blue for Large–McWilliams–Doney (LMD), and red for Mellor-Yamada 2.5 (MY25). The label ONLINE refers to the coupled simulation, while the labels "OFF (x1)", "OFF (x3)", "OFF (x5)", and "OFF (x10)" indicate the respective offline simulations with the time steps multiples used.

**Figure 12** presents time series of the biogeochemical tracers in the surface layer, comparing results from simulations using different DTs in the GLS mixing scheme configuration. During the first five months of the simulations, the offline-online differences exhibit higher variability, likely due to the model's spin-up period (shaded in blue). In this period, all DT simulations (x1, x3, x5, and x10) produce nearly identical results across variables, particularly for $O_2$ and CHL.

Then, two distinct transitions can be identified: the first, in pink shade, occurs around the middle of the 5th month (April), and the second, occurs about the 9th month (August). At the first transition, the bias variability stabilizes and becomes more consistent over time, marking the apparent end of the spin-up period. The second transition, in green shade, is characterized by reduced differences for $NO_3$, $NH_4$, $PO_4$, and $O_2$, alongside a noticeable increase in the difference for CHL (**Fig. 12**).

A peculiar artifact appears within the shaded yellow region of the plot, where errors in $PO_4$ unexpectedly diverge at DT x10 before converging with $O_2$ values. This divergence may represent a computational issue or the model's response to external forcing.

Regarding the effect of varying DT values, there is no consistent correlation between increasing DT and higher error magnitude. Instead, the extent of the differences appears variable-dependent and influenced by the simulation phase. After the spin-up period, differences between simulations propagate differently across tracers, especially for nutrients such as $PO_4$ and $NO_3$.

In the bottom layer (**Fig. S7**), a similar pattern emerges. However, in this case, the offline-online differences consistently increase with larger DT values. Despite these variations, the magnitude of differences in both surface and bottom layers remains within acceptable error margins for all variables.

We note that the four periods highlighted in **Fig. 12** and **Fig. S7** represent one possible interpretation of the observed temporal variability, based on apparent shifts in similarity across cases. These divisions may reflect transient behaviors such as spin-up dynamics or offline initialization effects; however, they could also be shaped by seasonal variability in the external forcings. Further investigation would be required to disentangle these factors.

Evaluating the effect of different DT values on vertical profiles of key biogeochemical tracers reveals that discrepancies between the results are minimal across all variables, with varying patterns across different profiles (**Fig. 13**). At Point 3, discrepancies between the results are almost negligible across all variables, indicating that there are basically no differences between the DTs. This suggests a high level of consistency in the model outputs at this location. This is of particular relevance since it is located near the Mississippi River mouth (**Fig. 1**).

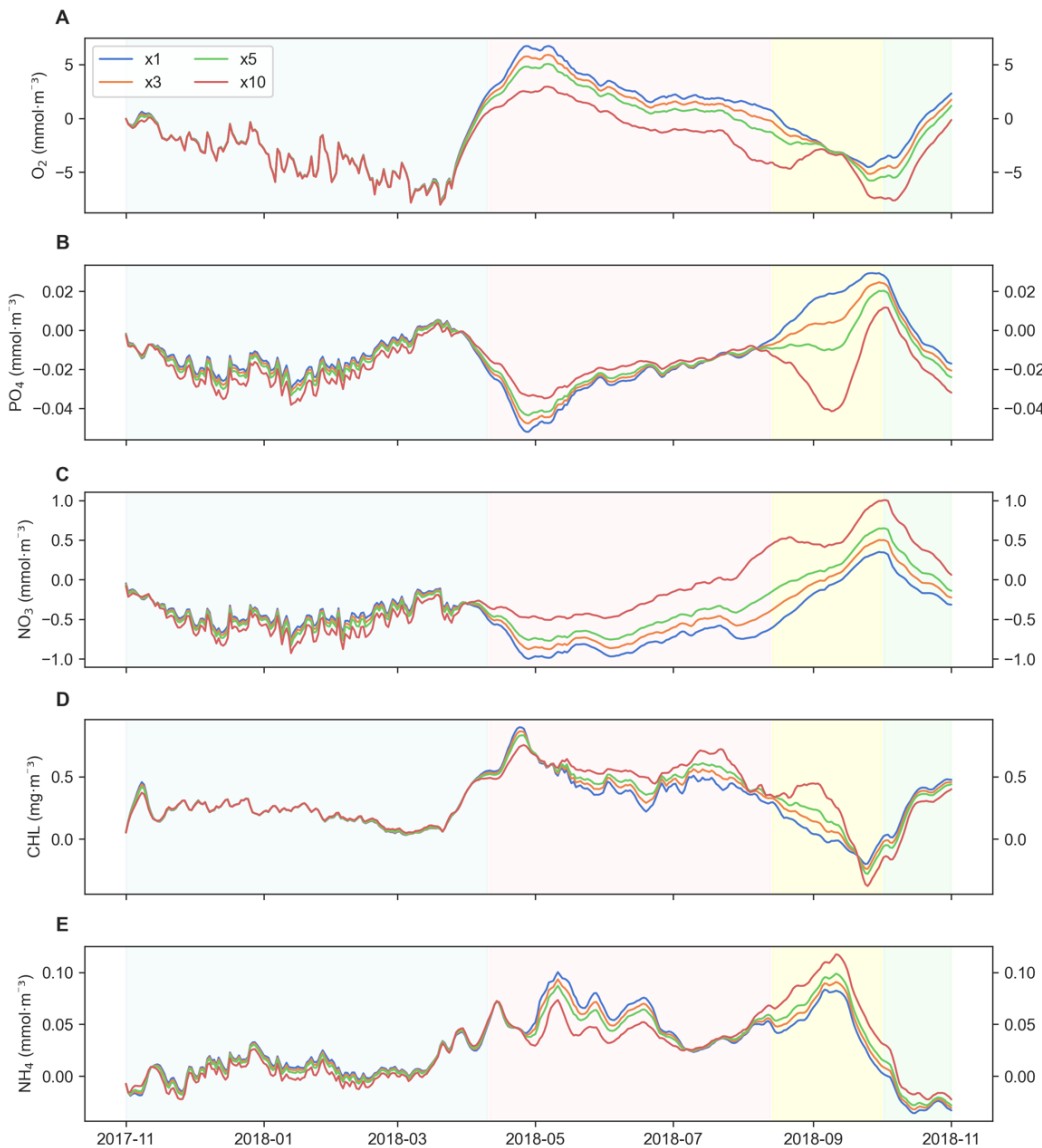

**Figure 12: Differences in simulation results at the surface layer across time steps (DTs) used in simulations with the Generic Length Scale (GLS) mixing scheme.** Panels show the following variables: (A) dissolved oxygen ($O_2$), (B) phosphate ($PO_4$), (C) nitrate ($NO_3$), (D) chlorophyll (CHL), and (E) ammonium ($NH_4$). The blue, orange, green, and red lines represent offline simulations with DT multiples of x1, x3, x5, and x10, respectively. The blue shaded area indicates the spin-up period where all simulations show high variability but converge to similar values. The pink shaded region marks the stabilization phase, with variables showing reduced variability and the emergence of differences between DT simulations. The yellow shaded region highlights a potential computational artifact or response to forcing, characterized by divergence in $PO_4$ and convergence in $O_2$. The green shaded region suggests seasonal dynamics, with a distinct change in behavior for the simulated variables.

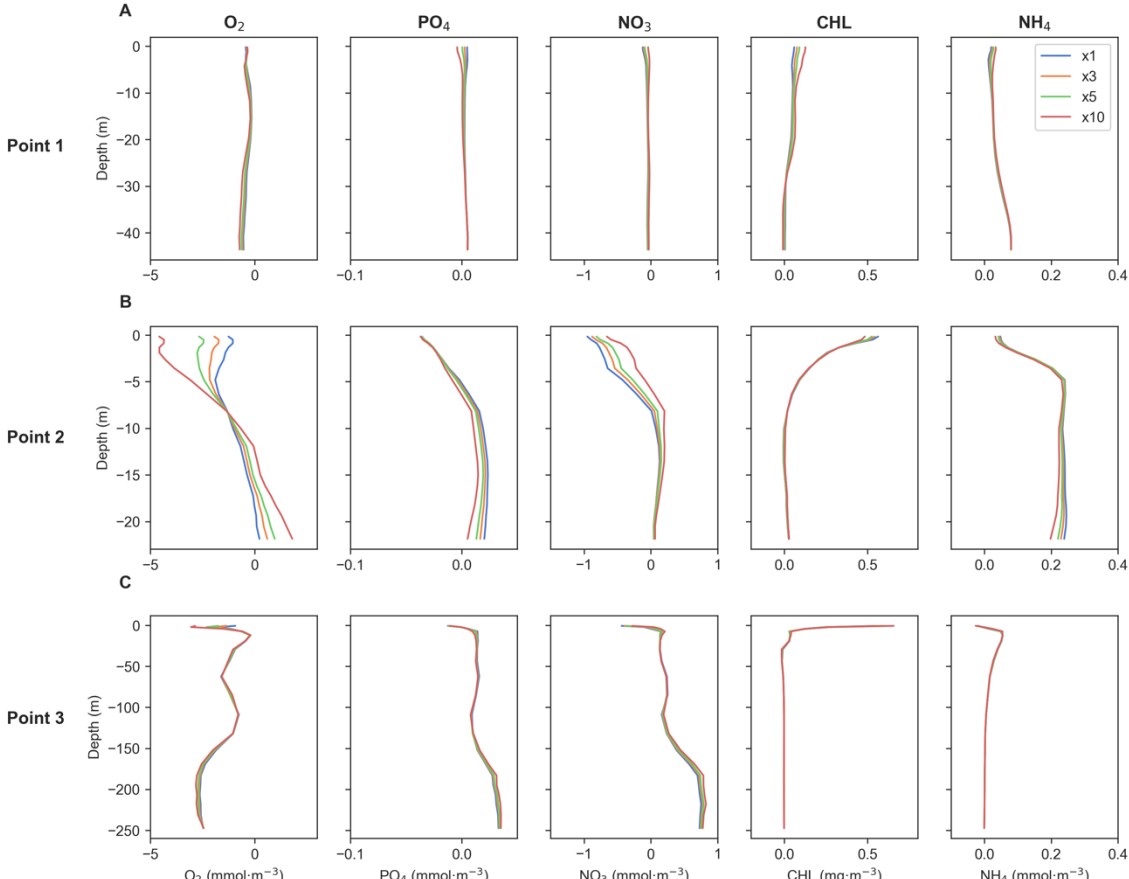

**Figure 13: Time-averaged vertical differences between offline and online simulations for the Generic Length Scale (GLS) configuration, based on the time steps (DTs) used.** Panels display data for: (A) Point 1, located offshore in the western part of the study area; (B) Point 2, near the mouth of the Atchafalaya River; and (C) Point 3, near the mouth of the Mississippi River. Refer to Figure 1 for specific point locations. The blue, orange, green, and red lines represent offline simulations with DTs multiples of x1, x3, x5, and x10, respectively, as shown in the legend in the upper right plot.

At Point 1, differences are slight and primarily localized in the upper meters of the water column. For all tracers the variations are minor, indicating that the choice of DT has a negligible impact on the results at this point. In contrast, Point 2 exhibits more dynamic behavior, with noticeable discrepancies both near the surface and at the bottom layers. Here, the impacts of the DT vary depending on the variable being examined.

Overall, despite localized variations, the findings from **Section 3.2** highlight that the computational efficiency gains from using larger DTs come at practically no cost to model accuracy. The differences between results for different DTs are minimal, reinforcing the robustness and reliability of the simulations across all examined variables.

## 4 Discussion and conclusions

In this study, we introduce and evaluate the Offline Fennel model, developed as an alternative to fully coupled physical-biogeochemical models. This model was tested in the NGoM against online (coupled) simulations using multiple mixing schemes —GLS, LMD, and MY25— to assess its performance in simulating key biogeochemical tracers such as $NO_3$, $NH_4$, $PO_4$, CHL, and $O_2$. The goal of this comparison is to determine whether the offline biogeochemical model can accurately replicate the results of coupled simulations within ROMS while offering substantial computational efficiency gains.

Our comparison reveals that the Offline Fennel model delivers high accuracy in reproducing the key biogeochemical tracers, with an average SS of 93% across all simulations. This demonstrates the model's strong ability to match the online coupled configuration, particularly in terms of time-series, spatial patterns, and vertical profiles. While the Offline Fennel model closely replicates the performance of the coupled system for most variables, some slight discrepancies were observed, particularly for $NH_4$. These discrepancies, while small, were located near coastal and river mouth regions (Atchafalaya and Mississippi rivers), areas that are inherently more challenging to model due to complex river-induced nutrient dynamics and mixing in shallow waters (Laurent et al., 2017). In addition, $NH_4$ is a short-lived tracer (Klawonn et al., 2019) that undergoes rapid transformations such as nitrification (oxidation of $NH_4$ to $NO_3$), and organic matter reaching the sediment is instantaneously remineralized, accounting for the loss of fixed nitrogen through denitrification (Fennel et al., 2013). This fast turnover makes $NH_4$ particularly sensitive to the temporal resolution of offline forcing fields, especially in coastal and riverine regions characterized by strong biogeochemical gradients (Yu et al., 2016; Laurent et al., 2017). However, this is not necessarily a limitation of the offline model but rather reflects the inherent difficulty in simulating highly dynamic environments and rapidly cycling tracers. In contrast, other tracers such as $NO_3$ and CHL, while also involved in dynamic processes (Fennel and Laurent, 2018), generally exhibit smoother spatiotemporal variability and are less sensitive to short-term fluctuations.

The inclusion of near-coastal points closer to river mouths, such as points 2 and 3 (**Fig. 1**), was intentional, as these locations represent areas with significant challenges for model validation. The fact that the Offline Fennel model performs well even in these difficult regions further emphasizes the robustness of the model for more general applications.

Despite these minor discrepancies, the model demonstrates strong performance, with $O_2$ showing particularly robust results, which is particularly notable given its importance in the NGoM ecosystem (Rabalais et al., 2002) and globally. This suggests that the Offline Fennel model is highly capable of reproducing both the spatial distribution and temporal evolution of biogeochemical tracers with minimal error.

The computational efficiency of the Offline Fennel model is one of its most significant advantages. By increasing the time-step by a factor of 10, we were able to reduce computational time from an average of 5 hours and 15 minutes to just 30 minutes,

which represents an 87% reduction in computational time. This drastic improvement is crucial for long-term simulations and large-scale applications, where running fully coupled models would be computationally limiting. The reduced computational time also opens the door for scenarios where multiple model runs are needed, such as sensitivity analyses, parameter tuning, or data assimilation approaches (Fennel et al., 2022), without the necessity of rerunning the hydrodynamic component each time. This feature greatly accelerates the simulation process and allows for broader exploration of different biogeochemical conditions at a fraction of the computational cost.

Furthermore, we observed that increasing the DT had practically no impact on the model results, which is highly relevant. While increasing the DT significantly reduces computational time, the discrepancies between simulations using different DTs were minimal. This finding further highlights the efficiency of the Offline Fennel model, as it allows for a substantial reduction in computational time without compromising model accuracy. This demonstrates that increasing the DT can be a viable strategy for accelerating simulations, particularly in large-scale or long-term studies, without introducing significant errors.

When comparing the three mixing schemes (GLS, LMD, and MY25), the GLS scheme consistently provided the best performance in terms of accuracy. The MY25 scheme showed similar results but was slightly less accurate compared to GLS. The LMD scheme showed the lowest performance overall. These differences are likely due to the fact that the GLS scheme, by design, is better at capturing small-scale turbulence and mixing processes, which are critical for accurate biogeochemical simulation in the NGoM. Although both MY25 and GLS simulations incorporated additional coefficients ('AK', 'gls', 'tke'), the performance for MY25 was not as robust as GLS. These results suggest that while incorporating additional coefficients can improve model accuracy, they are not necessarily a critical factor for this study. The relatively small differences between the three mixing schemes further highlight the robustness of the Offline Fennel model, which was able to handle a variety of mixing configurations without a significant loss of accuracy.

Although the Offline Fennel model demonstrated strong overall performance, some limitations remain. Testing additional tracer schemes or refining the model configuration for the NGoM could potentially address these small discrepancies. Furthermore, extending the model's application to other regions could offer further validation and highlight any region-specific limitations, though this is something to explore in future work. However, overall, the model has proven to be an effective tool for biogeochemical simulations in the NGoM.

In addition, a potential avenue for future study would be to repeat the one-year simulation in back-to-back cycles to better determine whether early-period variability reflects model spin-up effects or is fully driven by external forcing. Such an approach would help isolate transient initialization behaviours from recurring seasonal signals.

A particularly intriguing aspect of the results is the lack of growing error over time in the Offline Fennel model simulations, as typically observed in other studies, like data assimilation schemes (Berry and Harlim, 2017). This may partly reflect the relatively short simulation period (one year), the model domain, and the stabilizing influence of boundary conditions such as

prescribed riverine fluxes. However, it also suggests that biological variability remains well constrained by the physical state and that the simplification of feedback mechanisms in the offline framework does not result in significant error accumulation (Béal et al., 2010). Further investigation is needed to explore this phenomenon more thoroughly and assess whether this stability persists across different configurations and longer timeframes.

Finally, discrepancies between the offline and online simulations were smaller than those typically observed between different mixing schemes within the same application. This suggests that the choice of mixing scheme has a more significant impact on model accuracy than the distinction between offline and coupled configurations. Therefore, the offline model offers substantial benefits in terms of computational efficiency without compromising its ability to represent biogeochemical processes accurately.

Although the focus of this work is on the performance of the offline biogeochemical configuration, it is worth noting that different mixing (GLS, MY25, LMD) lead to notable differences in the predicted distributions of biogeochemical variables (e.g., Fig. 6). These differences highlight the sensitivity of biogeochemical tracer simulations to physical mixing processes, especially in coastal and shelf regions. A systematic evaluation of the biogeochemical impacts of turbulence parameterizations

would be an important direction for future research.

An important consideration when using offline biogeochemical models is the temporal resolution of the hydrodynamic archive used to force tracer transport (Thyng et al, 2021). In our study, all offline simulations were forced with 3-hourly velocity, temperature, salinity, and other physical fields extracted from the online model. This relatively fine sampling helps retain much

of the variability relevant to tracer advection and diffusion. Since all simulations use the same archived fields, the differences observed between the DTs cases are not due to changes in the input currents themselves, which are identical, but rather to the internal numerical treatment of tracer advection within the offline model.

Nevertheless, it is important to note that the hydrodynamic sampling rate used to force the offline model may limit the ability

of Offline Fennel to accurately reproduce biogeochemical tracer fields, particularly in dynamically active regions where currents fluctuate significantly on tidal or sub-daily timescales. While this was not a dominant source of error in our study area, the effect could become more pronounced in highly physically variable regions, such as strongly tidal or wind-driven coastal environments. In such cases, the loss of high-frequency momentum variability between archive snapshots could degrade the fidelity of biogeochemical tracer simulations. This highlights the importance of choosing an appropriate archive

sampling rate when configuring offline models, particularly for regions with energetic sub-daily dynamics.

In conclusion, the Offline Fennel model offers a promising alternative to coupled simulations, particularly in settings where computational resources are limited or when large-scale, long-term simulations are needed. The model accurately represents key biogeochemical processes, such as nutrient cycling, primary production, and oxygen dynamics in the NGoM, and its ability to drastically reduce computational time while maintaining high accuracy offers significant advantages for future applications. Furthermore, the model's autonomy from hydrodynamic processes also minimizes dependencies, providing flexibility in conducting extensive parameter tuning and sensitivity testing. While there are minor areas for improvement, the Offline Fennel model stands as a valuable tool for researchers working with ROMS hydrodynamic outputs or new ROMS configurations. While the results are specific to our model setup and the dynamics of the NGoM, they provide valuable insights for researchers with similar configurations and offer general guidelines for further applications.

## Appendix A. Explanation of code changes

This section describes the modifications made to Thyng et al. (2021) offline main code (https://github.com/kthyng/COAWST-ROMS-OIL/tree/master/ROMS) to make the offline biogeochemical model, namely Offline Fennel, working properly by acquiring the required forced variables, and to add new tracers and parameters. The most significant change, compared to the previous version, is the addition and operability of the following biological tracers: phosphate (PO4), river carbon detritus (RDeC) and river nitrogen detritus (RDeN). The following repository contains the modifications made to Thyng et al. (2021) original code: https://github.com/jcrespinesteve/OfflineFennel.

The new tracers and their respective equations have been incorporated to the "fennel.h" file. The offline model now includes phosphate (enabled via the PO4 C-preprocessing option), and river detritus computations for both nitrogen and carbon (enabled via the RIVER_DON option). It is important to note that river inputs, such as freshwater discharge and associated biogeochemical tracers, are prescribed in the model, rather than dynamically simulated. However, when RIVER_DON is activated, the model performs additional internal computations to represent the transformation of non-sinking dissolved organic matter from river sources (Yu et al., 2015). A similar approach is applied when PO4 is activated: phosphate is not simply added as a biogeochemical tracer, but instead integrated into the model's biogeochemical cycles through additional terms and modified equations following Laurent et al., (2017) Fennel model version, and ensuring consistency with the rest of the nutrient dynamics. In addition, another oxygen computation, RW1_OXYGEN_SC, has been added to the file. When this C-processing option is activated, the biogeochemical model uses Wanninkhof (2014) air-sea flux parameterization to calculate oxygen values.

The new tracers and their respective equations have been incorporated into the "fennel.h" file. The offline model now includes phosphate (enabled via the PO4 C-preprocessing option), and the capability to handle riverine dissolved organic nitrogen and

carbon (enabled via the RIVER_DON option). It is important to note that river inputs—such as freshwater discharge and associated biogeochemical tracers—are prescribed in the model, rather than dynamically simulated. However, when RIVER_DON is activated, the model performs additional internal computations to represent the transformation of non-sinking dissolved organic matter from river sources. For instance, the prescribed remineralization rate (RDeRRN) is used to modify the concentration of riverine DON and convert it into ammonium within the model. This functionality was adapted from Thyng

et al. (2021) and integrated into the offline model. Additionally, a new oxygen computation scheme (RW1_OXYGEN_SC) has been added. When this option is activated, the biogeochemical model uses the air-sea gas exchange parameterization from Wanninkhof (2014) to calculate oxygen fluxes.

Metadata indices for the new variables are now included in the "fennel_var.h" file. The "fennel_def.h" file incorporates new

input parameters to calculate the following tracers: phyto-phosphate:nitrogen ratio ('R_P2N'), inverse half saturation for phytoplankton phosphate uptake ('K_PO4'), remineralization rate for nitrogen and carbon river detritus ('RDeRRN' and 'RDeRRC'). Such new input parameters are likewise defined in the "fennel_inp.h", "fennel_mod.h", and "fennel_wrt.h" files, so the new parameters are read, allocated and written out in the offline simulations.

To facilitate the configuration process for offline simulations, some changes have been also made in the "globaldefs.F" module. Now, when OFFLINE and OFFLINE_BIOLOGY are defined in the header file, ATCLIMATOLOGY is automatically activated for processing and allocating the active tracers of the simulation, which are temperature and salinity. Moreover, the "checkvars.F" file was also revised to activate the acquisition of the active tracers during offline simulations. Specifically, this file now ensures that when the OFFLINE and ATCLIMATOLOGY options are defined, the model correctly identifies and

retrieves the active tracers. This is indispensable for the Offline Fennel model, as temperature is used for light limitation to compute phytoplankton growth, and salinity is needed to calculate oxygen saturation.

In the "checkdefs.F" file, new settings for verifying C-processing options have been incorporated. These settings include the addition of PO4 dynamics (PO4), the river detritus equation (RIVER_DON), and the inclusion of river biology point sources

(RIVER_BIOLOGY).

Finally, the "set_data.F" file was modified to prevent the model from accessing subsequent time step values of sea surface height and 3D momentum climatologies. This adjustment was necessary to eliminate a shift that occurred when processing climatology fields in the bottom layers, which had previously propagated a bias toward the surface when calculating

biogeochemical tracer concentrations.

**Appendix B. Offline Fennel guide**

To conduct offline biogeochemical simulations using Offline Fennel, users have to run a specific version of COAWST/ROMS available in https://github.com/kthyng/COAWST-ROMS-OIL. Requirements and considerations for setting up biogeochemical offline simulations in ROMS using Offline Fennel are provided below.

**Climatology and forcing files**

Input the hydrodynamic model outputs as the climatology forcing ('CLMNAME') of the model. The variables needed for the climatology files, with ROMS required dimensions between brackets, are:

- Free-surface ('zeta') *(time, eta_rho, xi_rho)*
- Vertically integrated u-momentum component ('ubar') *(time, eta_u, xi_u)*
- Vertically integrated v-momentum component ('vbar') *(time, eta_v, xi_v)*
- u-momentum component ('u') *(time, s_rho, eta_u, xi_u)*
- v-momentum component ('v') *(time, s_rho, eta_v, xi_v)*
- S-coordinate vertical momentum component ('omega') *(time, s_w, eta_rho, xi_rho)*
- Temperature *(time, s_rho, eta_rho, xi_rho)*
- Salinity *(time, s_rho, eta_rho, xi_rho)*
- Solar shortwave radiation ('swrad') *(time, eta_rho, xi_rho)*
- Surface net heat flux ('shflux') *(time, eta_rho, xi_rho)*
- AKv, AKt, Aks optional (depending on mixing scheme) *(time, s_w, eta_rho, xi_rho)*
- Tke optional (depending on mixing scheme) *(time, s_w, eta_rho, xi_rho)*
- GLS optional (depending on mixing scheme) *(time, s_w, eta_rho, xi_rho)*

For the offline forcing file (FRCNAME), with ROMS required dimensions between brackets, the variables needed for the climatology files are:

- Solar shortwave radiation ('swrad') *(time, eta_rho, xi_rho)* [WARNING: Offline Fennel is very sensitive to this variable.]
- Surface net heat flux ('shflux') *(time, eta_rho, xi_rho)*
- Surface u-momentum stress ('sustr') *(time, eta_u, xi_u)*
- Surface v-momentum stress ('svstr') *(time, eta_v, xi_v)*

**Header file**

- Define OFFLINE and OFFLINE_BIOLOGY flags to conduct offline biogeochemical simulations.
- Do not define BULK_FLUXES, SOLAR_SOURCE nor DIURNAL_SRFLUX flags, since all forcing comes from the hydrodynamic model outputs to be introduced in the climatology and forcing files.
- Define ATCLIMATOLOGY to process and allocate active tracers (T and S). This is fundamental for phytoplankton growth and oxygen computation in the biogeochemical model.
- Define OCLIMATOLOGY for processing the variable 'omega' provided in the climatology forcing file.
- For best accuracy, use the same tracer advection scheme as the physical run. Use TS_MPDATA for best tracer advection results (Thyng et al., 2021).
- Use OUT_DOUBLE and PERFECT_RESTART for best results.
- Define MIX_CLIMATOLOGY to use Tke and GLS, and AKXCLIMATOLOGY for akt, aks, and akv use.

**Configuration file**

- A multiplier of the hydrodynamic time step is a good option for the offline simulation time step DT. The present study found that a DT equal to 1, 3, 5, and 10 times the physical DT provided good results. However, some testing for the implementation setup is recommended. WARNING: Note also that the offline time step must be proportional to the hydrodynamic output frequency, and that it cannot be larger than the latter.
- Close all boundaries for the physics, since all data comes from the hydrodynamic model outputs and an open boundary would modify the hydrodynamics.
- Turn on "LsshCLM", "Lm2CLM", "Lm3CLM" and "LtracerCLM" to process the climatology forcing file.
- Do not activate climatology nudging ("LnudgeM2CLM", "LnudgeM3CLM", "LnudgeTCLM"), since the physical output must be entirely forced.
- Do not activate tracers for sources ("LtracerSrc"), since this has already been computed in the physical simulation.
- Turn on the momentum for Sources/Sinks if river nutrients (RIVER_BIOLOGY) are to be added. This will not modify the hydrodynamics of the model, as it will only impact the biology and nutrients of the model. If "LuvSrc" or "LwSrc" are not activated, no nutrients will come out from the river points.
- A specific varinfo.dat file available here (https://github.com/kthyng/COAWST-ROMS-OIL/blob/master/ROMS/External/varinfo-offline.dat - last access: 14 Apr 2023) must be used for offline simulations, as it has been modified to include the additional variables for the offline model. The latter adjustment enables the offline input of the physical result to be input as climatology without undergoing file processing to rename variable attributes.

**Acknowledgments**

The authors would like to express their gratitude to Prof. Fennel and Dr. Laurent for their guidance, as well as for providing the necessary files and configuration for the model implementation in the NGoM. Their support during the summer of 2022, when JC was hosted in their research group, was instrumental in integrating the new tracers and parameters into the offline biogeochemical model Offline Fennel.

JC is supported by a grant for the recruitment of researchers in training (FI-SDUR) from the Catalan Government (Generalitat de Catalunya). The CRG Marine Geosciences group is funded by the Catalan Government under its excellence research groups program (ref. 2021 SGR 01195) (JC, MC). JS is supported by grant CEX2019-000928-S, funded by the Agencia Española de Investigación (AEI) (10.13039/501100011033). The research has been funded under: 'Fons Climàtic Llei 4/2017, del 28 de

març de la Generalitat de Catalunya'.

MC acknowledges the support of Tecnoambiente for the Sustainable Blue Economy Chair at the University of Barcelona. Additionally, JS acknowledges the Catalan Government (Generalitat de Catalunya) for the contract PYMEDEASCAT, related to the project "Prospectiva d'emissions a Catalunya: pymedeascat_pro".

**Author contributions**

JC developed the offline biogeochemical model, conducted both online and offline simulations, generated all figures, performed the data analysis, and wrote and edited the manuscript. JS contributed to the analysis of the results and edited and reviewed the manuscript. MC also participated in the analysis of the results and contributed to the editing and review of the

manuscript.

**Competing interests**

Authors declare that they have no competing interests.

**Code and Data availability**

The current version of the Offline Fennel model is available from the project website: https://github.com/jcrespinesteve/OfflineFennel under the MIT licence. The exact version of the model used to produce the results presented in this paper is archived on Zenodo (Crespin, 2025a; doi: 10.5281/zenodo.14916223). Additionally, the input data and scripts to run the model for all the simulations discussed in this paper, along with the processed outputs, are also

available (Crespin, 2025b; doi: 10.5281/zenodo.14930138).

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
