# Peer review of "Offline Fennel: A High-Performance and Computationally Efficient Biogeochemical Model within the Regional Ocean Modeling System (ROMS)"

_EGUsphere, 2025_

## Author Response (AR1)

**RESPONSES TO REVIEWERS**

We sincerely appreciate the time and effort the reviewers have dedicated to our manuscript. Their constructive feedback has been invaluable in refining our study and improving the clarity of our arguments. Below, starting with Rev. 1 and ending with Rev. 2, we address each of the reviewer's comments and suggestions in detail and describe the corresponding changes made to the manuscript.

**REVIEWER 1**

The paper describes the details of the biogeochemical Fennel model for ROMS that can be run in an offline model. The aim of the paper is to report on the model's capability to reproduce the biogeochemical fields in comparison to an online model and to report on the model's efficiency. The validation of the model itself is outside of the scope. The links to a GitHub page and detailed instructions to the potential users of the model are also provided in the Appendices to the paper.

The aim of the paper is achieved, and the paper reads very well and is well structured. English is impeccable and I really enjoyed reading this paper. I particularly welcome the detailed instructions contained in the Appendices, and I would like to congratulate the authors on this development and on putting together a solid and very well written manuscript. Being a biogeochemical modeler myself I thoroughly understand the need for running the biogeochemical models in an offline mode and all computational savings this approach offers. It has been known for years that the biogeochemical models do not need to run with the same time stepping as the hydrodynamic models and yet few developments tool place so far in this regard.

**RESPONSE:** Thank you for your thoughtful and detailed review of our manuscript. We appreciate your positive feedback regarding our paper and are especially pleased to hear that the Appendices and the practical information for users is appreciated. It is encouraging to receive this feedback from a fellow biogeochemical modeler who understands the challenges and motivations behind developing an efficient offline biogeochemical framework. We agree that, despite long-standing awareness of the computational advantages of offline approaches, relatively few tools have been implemented and documented for community use. Our aim was to contribute toward filling that gap, and we are grateful for the reviewer's recognition of this effort.

There are only some minor deficiencies that it would be good to see addressed by the authors.

**RESPONSE:** We have carefully addressed all of these in the revised manuscript. A detailed point-by-point response to each suggestion is provided below.

Ln 88 and the paragraph starting ln 91: I did not follow relevant literature, but the changes to the original Fennel model (2006, 2008) reported in this sentence, in my view, are likely to significantly alter the solution that would be achieved with the original Fennel model. I note the modifications were published in the papers by Laurent et al. (2012), Fennel et al., 2013, Yu et al., 2015 and Laurent et al. 2017. Is this model still called

the Fennel model in the literature? The original Fennel model does not contain the phosphates and has only two detrital pools

**RESPONSE:** The model has indeed evolved since its original formulation in Fennel et al. (2006, 2008), with modifications reported in later studies such as Laurent et al. (2012), Fennel et al. (2013), and Yu et al. (2015). However, the current configuration (including the addition of phosphate, oxygen, and riverine detritus) continues to be referred to as the Fennel model in the literature (e.g., Laurent et al., 2017; Fennel et al., 2013; Yu et al., 2015). This naming convention is also consistent within the ROMS community and its documentation, as shown by references to 'bio_Fennel.in' and 'fennel.h' in the ROMS source code, and on the official ROMS wiki (www.myroms.org/wiki/bio_Fennel.in).

However, to prevent any potential confusion for readers who may be unfamiliar with this model, we have revised the manuscript to clarify this distinction as follows (changes with respect to the initial version in **bold**):

**L97-99:** *The **version of the biogeochemical model used in this study** builds on the ROMS biogeochemical component developed by Fennel et al. (2006, 2008) and **later expanded to account for phosphate (Laurent et al., 2012), oxygen (Fennel et al., 2013), and non-sinking river detritus (Yu et al., 2015b). The current biogeochemical model Fennel includes 15 state variables: […]***

It is also worth noting that the version of the Fennel model used in the offline framework developed by Thyng et al. (2021) did not include these more recent biogeochemical extensions, and it was not functional as an offline biogeochemical model. For this reason, and to ensure consistency with the up-to-date version of the Fennel model, we implemented modifications to the offline module. These adjustments were necessary for proper functioning and are described in the Methods section and with more detail in Appendix A.

Ln 140 Is surface net heat flux used in the biogeochemical model? How? Solar shortwave radiation provides the light intensity for the growth equation, but what is the net heat flux used for?

Ln 141 Same question as above for the u- and v- momentum stress

**RESPONSE**: In our offline framework, all hydrodynamics – including velocity fields (u and v), temperature, and salinity –, are prescribed from the online parent model and are not actively computed during the offline simulation. The biogeochemical tracers are advected and mixed based on this prescribed physical forcing. Consequently, u- and v-momentum fields are essential for the correct advection of tracers and must be included in the offline archive.

Regarding the net surface heat flux, while it does not directly influence any biogeochemical source/sink terms in the current biogeochemical model code. It was included in the archive for consistency with the structure proposed by Thyng et al. (2021) and to preserve flexibility. For instance, surface heat flux could be relevant for future implementations where temperature-dependent biogeochemical processes are

included or when using bulk formulations (e.g., air-sea gas exchange calculations). Therefore, these variables contribute to a more comprehensive and modular offline setup, while helping the future developments of the model by potential users.

Ln 145 This is a repetition of line 117

**RESPONSE:** We acknowledge the repetition and have reviewed the paragraph to eliminate redundancy as follows:

**L157:**  *A series of online (coupled) and offline (uncoupled) simulations….*

Ln 201 What weights were applied and what is n in equation (2)?

**RESPONSE**: We realize that our original wording may have suggested that non-uniform weighting was applied. We clarify that equal weights were used across the vertical levels in the RMSE calculation, and we have revised the text accordingly. Regarding the equation, the variable "n" represents the total number of vertical levels (*s_rho*). This has now been explicitly stated in the manuscript as follows:

**L215-222:** *In this study, RMSE was computed across the vertical coordinate dimension ('s_rho'), representing depth levels. **Equal weights were used across all vertical layers, meaning that each depth level contributed equally to the overall error. This approach avoids bias toward any specific layer and provides a balanced assessment of model performance across the entire water column.***

$$RMSE = \sqrt{\frac{1}{n}\sum_{i=1}^{n}(C_{on} - C_{off})^2} \qquad (Eq.\ 2)$$

*where $C_{off}$ and $C_{on}$ are the time-averaged concentrations of a tracer on the 3D grid for the offline and online simulations, respectively, **and n is the total number of vertical levels.***

Ln 335 It is reported that the biggest difference between the offline and online calculations is observed for NH4, but potential reasons are not discussed. I suggest a paragraph is added in the discussion section. Is it because it is short lived and undergoes relatively quick oxygenation to nitrates? If that was the reason, how about then the accuracy of the solution for other biogeochemical variables, e.g. NO3 and CHL during the times of rapid transformations, e.g. rapid blooms?

**RESPONSE**: We appreciate your suggestion to discuss the significant differences observed for $NH_4$. We have added a paragraph in the Discussion section addressing potential reasons for this discrepancy, including the rapid transformations of $NH_4$ and its implications for other biogeochemical variables during periods of rapid change. See below.

**L521-533:** *While the Offline Fennel model closely replicates the performance of the coupled system for most variables, some slight discrepancies were observed, particularly for $NH_4$. These discrepancies, while small, were located near coastal and river mouth regions (Atchafalaya and Mississippi rivers), areas that are inherently more challenging to model due to complex river-induced nutrient dynamics and mixing in shallow waters (Laurent et al., 2017). In addition, $NH_4$ is a short-lived tracer (Klawonn et al., 2019) that undergoes rapid transformations such as nitrification (oxidation of $NH_4$ to $NO_3$), and organic matter reaching the sediment is instantaneously remineralized, accounting for the loss of fixed nitrogen through denitrification (Fennel et al., 2013). This fast turnover makes $NH_4$ particularly sensitive to the temporal resolution of offline forcing fields, especially in coastal and riverine regions characterized by strong biogeochemical gradients (Yu et al., 2016; Laurent et al., 2017). However, this is not necessarily a limitation of the offline model but rather reflects the inherent difficulty in simulating highly dynamic environments and rapidly cycling tracers. In contrast, other tracers such as $NO_3$ and CHL, while also involved in dynamic processes (Fennel and Laurent, 2018), generally exhibit smoother spatiotemporal variability and are less sensitive to short-term fluctuations.*

Figure 5: I appreciate this is somewhat outside of the scope of the paper, but the differences in the predictions of individual biogeochemical variables appear to be very significant when using different turbulence schemes. Do the authors report on it in a separate manuscript? Is there a scope to include some discussion on it in this paper?

**RESPONSE:** We recognize the importance of the differences in predictions using various turbulence schemes. While the primary focus of this study is on evaluating the performance of the offline biogeochemical model, we agree that the impact of different turbulence closure schemes on biogeochemical tracer distributions could be indeed substantial and scientifically relevant. However, a thorough investigation of this topic is beyond the scope of the current paper and would require a dedicated analysis. We have now included a short paragraph in the Discussion to acknowledge this and suggest it as an avenue for future work.

**L594-598:** *Whereas the focus of this work is on the performance of the offline biogeochemical configuration, it is worth noting that different mixing (GLS, MY25, LMD) lead to notable differences in the predicted distributions of biogeochemical variables (e.g., Fig. 5). These differences highlight the sensitivity of biogeochemical tracer simulations to physical mixing processes, especially in coastal and shelf regions. A systematic evaluation of the biogeochemical impacts of turbulence parameterizations would be an important direction for future research.*

Ln 337 and Figure 6: What are the potential reasons for the GLS scheme resulting in least discrepancies between the online and offline simulations. Some comments on it would be useful in this manuscript.

**RESPONSE**: While all schemes perform similarly, GLS shows slightly better agreement, which may be related to the specific structure of the output fields available for the offline forcing. In particular, the GLS simulation includes additional physical fields such

as turbulent kinetic energy (TKE), the generic length scale (GLS), and vertical mixing coefficients (AKv, AKt, AKs), which may contribute to a more consistent offline representation of mixing and tracer transport. These additional fields may help better preserve vertical mixing characteristics in the offline model. In the initial submitted manuscript, the Discussion addressed this (L534-542, now 560-568):

**L560-568:** *"When comparing the three mixing schemes (GLS, LMD, and MY25), the GLS scheme consistently provided the best performance in terms of accuracy. The MY25 scheme showed similar results but was slightly less accurate compared to GLS. The LMD scheme showed the lowest performance overall. These differences are likely due to the fact that the GLS scheme, by design, is better at capturing small-scale turbulence and mixing processes, which are critical for accurate biogeochemical simulation in the NGoM. Although both MY25 and GLS simulations incorporated additional coefficients ('AK', 'gls', 'tke'), the performance for MY25 was not as robust as GLS. These results suggest that while incorporating additional coefficients can improve model accuracy, they are not necessarily a critical factor for this study. The relatively small differences between the three mixing schemes further highlight the robustness of the Offline Fennel model, which was able to handle a variety of mixing configurations without a significant loss of accuracy."*

Figure 6: I suggest it would be more informative if the differences here were reported in the percentage terms.

**RESPONSE:** We appreciate the reviewer's suggestion to report differences in percentage terms. While we agree that percent differences can be visually informative in some contexts, we believe this representation could be misleading for some biogeochemical tracers – particularly in areas where concentrations are extremely low.

In such regions, even tiny absolute differences (e.g., < 0.05 mmol·m⁻³) can yield large percentage values (e.g., >30%), despite being negligible in magnitude. This results in a visual exaggeration of errors. For example, what appears as a near-zero difference (gray) in the absolute scale may misleadingly appear as a significant deviation in percentage terms, giving the impression of a much greater error magnitude than is actually the case.

For this reason, we chose to present absolute differences in Fig. 6, as they provide a more balanced and meaningful depiction of model performance across all tracers.

Ln 431 '…, but the time allocated for file writing was insufficient' – I do not understand this statement. Couldn't this time be increased? What was the exact reason of the failure of x15 option? I was of an impression from the earlier parts of the manuscript that it had to do with model stability.

**RESPONSE:** The model successfully generated the log file, but the output files could not be written. While we initially suspected that the problem might be related to model stability, the model actually ran without any apparent instability during the simulation. This suggests that the issue is likely related to the output file writing process itself rather than the overall stability of the model.

Consequently, and in agreement with Reviewer 2's related comment, we have removed the x15 case from Fig. 10 to avoid potential confusion about its validity. However, we have retained it in Table S1 for completeness, as it helps document the limits of the tested DT configurations explained in the Methods section.

Discussion section: The discussion seems to be missing an important aspect: how is the accuracy of the biogeochemical model affected by the sampling rate of the hydrodynamic archive? Are there differences in the currents when the model is running with DT x1 vs DT x10? The differences in the biogeochemical variables under different DTs may be due to the discrepancies in advection due to the sampling rates of the hydrodynamic archive. This will become even more pronounced in highly tidal regions and/or coastal locations under strong influence of the wind.

**RESPONSE:** In our study, the hydrodynamic archives used to force the offline biogeochemical model were sampled at a fixed temporal resolution of 3 hours across all scenarios (cf. Methods). The reviewer is correct in noting that using a temporally coarser sampling rate in the hydrodynamic archive (e.g., longer intervals between forcing hydrodynamic fields) could introduce discrepancies in the representation of advection and mixing processes, particularly in regions with strong tidal currents or wind forcing.

In our case, while the archive's temporal resolution already is relatively fine (3-hourly), any interpolation or time-stepping scheme within the offline model could still lead to some smoothing or loss of high-frequency variability —especially if biogeochemical tracers are sensitive to short-term fluctuations in circulation.

We have now added a paragraph in the Discussion section to explicitly address this point, acknowledging the potential limitations of offline coupling due to the finite sampling frequency of the archived fields and how this may affect the accuracy of tracer advection. This consideration is of relevance for future applications in more dynamically active regions or when using coarser temporal resolutions in the archive.

**L600-613:** *A relevant consideration when using offline biogeochemical models is the temporal resolution of the hydrodynamic archive used to force tracer transport (Thyng et al., 2021). In our setup, all offline simulations were forced with 3-hourly velocity, temperature, salinity, and other physical fields extracted from the online model. This relatively fine sampling helps retaining much of the variability related to tracer advection and diffusion. Since all simulations use the same archived fields, the differences observed between the DTs cases are not due to changes in the input currents themselves, which are identical, but rather to the internal numerical treatment of tracer advection within the offline model.*

*Nevertheless, it should be noted that the hydrodynamic sampling rate used to force the offline model may limit the ability of Offline Fennel to accurately reproduce biogeochemical tracer fields. While this was not a dominant source of error in our study area, the effect could become more pronounced in highly physically variable regions, such as strongly tidal or wind-driven coastal environments. In such cases, the loss of high-frequency momentum variability between archive snapshots could degrade the fidelity of biogeochemical tracer simulations. This highlights the*

*importance of choosing an appropriate archive sampling rate when configuring offline models, particularly for regions with energetic sub-daily dynamics.*

Moreover, in the Methods section, we have specified:

**L151-155:** *Following the recommendation of Thyng et al. (2021), a 3-hourly hydrodynamic input frequency was selected to run the offline simulation. In their Gulf of Mexico simulations, they estimated an advection timescale of approximately 5.6 hours, based on a characteristic velocity of 0.5 m/s and length scale of ~10 km. Our choice of a 3-hour interval thus falls well below this threshold, providing at least one output every ~0.5 advection timescales (Thyng et al., 2021). This frequency is therefore adequate to resolve key physical changes and ensure accurate offline interpolation of biogeochemical tracers.*

Ln 584: 'river detritus computations' – is river modelled or is it just a forcing in the model? I thought river flows and river inputs are simply prescribed in the model, but this statement suggests that some sort of computations take place – see also line 605, a CPP flag RIVER_BIOLOGY.

**RESPONSE**: River inputs are prescribed in the model, and we have to ensure that the terminology used is consistent and clear. Specifically, we explain now that if "RIVER_DON" is activated, an additional biological tracer (or two if "CARBON" is defined) is included to represent non-sinking dissolved organic matter from rivers, as described in Yu et al. (2015). This clarification will help readers understand the role of river inputs in the biogeochemical model.

We clarify now in the manuscript that river flows and biogeochemical inputs are indeed prescribed in the model and not dynamically simulated. However, when the "RIVER_DON" CPP option is activated, additional internal computations are performed within the model to represent the transformation of non-sinking dissolved organic matter from river sources. For example, the evolution of river detritus (e.g., iRDeN) over time, with the remineralization rate transformation (RDeRRN), which was not included in Thyng et al. (2021) version of the biogeochemical model. This functionality is what we refer to when mentioning "river detritus computations". We have revised the text to make this distinction clear and ensure consistent terminology as follows:

**L633-643:** *The new tracers and their respective equations have been incorporated to the "fennel.h" file. The offline model now includes phosphate (**enabled via the** PO4 C-preprocessing option), and river detritus computations for both nitrogen and carbon (**enabled via the** RIVER_DON option). **It is important to note that river inputs, such as freshwater discharge and associated river biogeochemical tracers, are prescribed in the model. However, when RIVER_DON is activated, the model performs additional internal computations to represent the transformation of non-sinking dissolved organic matter from river sources (Yu et al., 2015). A similar approach is applied when PO4 is activated: phosphate is not simply added as a biogeochemical tracer, but instead integrated into the model's biogeochemical cycles through additional terms and modified equations following Laurent et al., (2017) Fennel model version, and***

*ensuring consistency with the rest of the nutrient dynamics. In addition, another oxygen computation, RW1_OXYGEN_SC, has been added to the file. When this C-processing option is activated, the biogeochemical model uses Wanninkhof (2014) air-sea flux parameterization to calculate oxygen values.*

Ln 641: add 'Surface'

**RESPONSE:** It is not just surface temperature. Please note that this is the climatology forcing from ROMS, which includes active tracers (temperature and salinity) in 4D (*ocean_time, s_rho, eta_rho, xi_rho*), so that it's not only the superficial dimension. For clarity, we have included the dimensions in the lists of climatology and forcing files for those who are not familiar with the dimensions required by ROMS.

**L687-711:** *Input the hydrodynamic model outputs as the climatology forcing ('CLMNAME') of the model. The variables needed for the climatology files**, with ROMS required dimensions between brackets,* are:*

- *Free surface ('zeta')* **(time, eta_rho, xi_rho)**
- *Vertically integrated u-momentum component ('ubar')* **(time, eta_u, xi_u)**
- *Vertically integrated v-momentum component ('vbar')* **(time, eta_v, xi_v)**
- *u-momentum component ('u')* **(time, s_rho, eta_u, xi_u)**
- *v-momentum component ('v')* **(time, s_rho, eta_v, xi_v)**
- *S-coordinate vertical momentum component ('omega')* **(time, s_w, eta_rho, xi_rho)**
- *Temperature* **(time, s_rho, eta_rho, xi_rho)**
- *Salinity* **(time, s_rho, eta_rho, xi_rho)**
- *Solar shortwave radiation ('swrad')* **(time, eta_rho, xi_rho)**
- *Surface net heat flux* **('shflux')** *(time, eta_rho, xi_rho)*
- *AKv, AKt, Aks optional (depending on mixing scheme)* **(time, s_w, eta_rho, xi_rho)**
- *Tke optional (depending on mixing scheme)* **(time, s_w, eta_rho, xi_rho)**
- *GLS optional (depending on mixing scheme)* **(time, s_w, eta_rho, xi_rho)**

*For the offline forcing file (FRCNAME),* **with ROMS required dimensions between brackets,** *the variables needed for the climatology files are:*

- *Solar shortwave radiation ('swrad')* **(time, eta_rho, xi_rho)** *[WARNING: Offline Fennel is very sensitive to this variable.]*
- *Surface net heat flux ('shflux')* **(time, eta_rho, xi_rho)**
- *Surface u-momentum stress* **('sustr')** *(time, eta_u, xi_u)*
- *Surface v-momentum stress* **('svstr')** *(time, eta_v, xi_v)*

Ln 662: 'and that it cannot be larger than the latter' – I suggest the word WARNING is added before this statement

**RESPONSE:** Done (see L730 in the revised manuscript).

**L730:** *WARNING: Note also that the offline time step must be proportional to the hydrodynamic output frequency, and that it cannot be larger than the latter.*

Once again, thank you for your valuable feedback. We believe that addressing all these points have significantly enhanced the quality of our manuscript.

**Reviewer 2**

GENERAL COMMENTS

The manuscript innovates over the work of Thyng et al. 2021 by providing two new valuable contributions: biogeochemical sinks/sources, and salinity/temperature as forcing variables. The results are reassuring and convincing, with differences between online/offline configurations being generally smaller than those associated with a change in mixing scheme. The manuscript provides a diverse set of visualizations (across time and spatial dimensions) demonstrating that the online/offline differences are generally unconcerning. The code is publicly available and I do see the relevance and usefulness of the study for the community at large and for the readers of GMD.

The criticisms I listed below (see "SPECIFIC COMMENTS" and "OTHER COMMENTS") are fairly common for a student/ECR paper, and they should be addressable within a normal "major revision" period. I do not have further concerns to report.

**RESPONSE:** We would like to thank Reviewer 2 for their positive and encouraging assessment of our manuscript. We sincerely appreciate your recognition of the contributions we have made beyond the work of Thyng et al. (2021). We have carefully considered your specific and other comments, and we have revised the manuscript accordingly to address each point. Detailed responses to each of those comments are provided below. We are confident that these revisions have strengthened the manuscript.

SPECIFIC COMMENTS

(1) In many instances, the authors over-emphasize differences in the ability of the 3 mixing schemes (GLS,MY25,LMD) to reproduce the "online" results.

For example...

- Lines 229-230,244: Are the GLS numbers *significantly* better than those from MY25 and LMD? When I look at Table 1, all these numbers are quite close...
- Line 267: GLS does not "consistently" outperform the others. In Figs.S1-S3, around March 2018, the red line (MY25) is above the green line (GLS). Considering that the 3 lines are one of top of the other for the majority of the time, I'm not convinced it's fair to say that one of them is "outperforming" the others.
- Line 280: It is hard for me to see in Fig.3 that "GLS continues to provide the best performance". They all look similar to my eyes.

- Lines 338-339: It is hard to see from Fig.6 that "smallest discrepancies are consistently found with the GLS mixing scheme". They look similar to me.
- Line 397: The LMD exhibiting "more variability in the fall" is hard to see in Fig.S5. They look similar to me.
- Line 402: GLS demonstrating "the smallest deviations" is very to see in Fig.S5. They look similar to me.
- Line 411: It's really hard for me to say from Fig.9 that GLS is the "most effective scheme". As far as I'm concerned, all 3 mixing schemes are doing similarly well based on Fig.9. And there is at least one instance (Fig.9A, DJF, near the surface) where we can say for certain that GLS is actually worse than the other two.

Overall, my recommendation would be to only emphasize in the text differences that are clear and evident from the figures/tables that you provide to the reader.

**RESPONSE:** We acknowledge that our original text may have overstated some of the differences in performance among the mixing schemes. While we intended to highlight some consistent, albeit subtle, patterns favoring GLS in certain metrics, we agree that these distinctions are often small and not always visually or statistically significant.

In response, we have revised the manuscript to adopt more neutral and cautious language throughout the relevant sections. Specifically:

- Lines 229–230, 244: We have removed claims suggesting GLS was "better" and instead described the results as being comparable across schemes, with only minor differences noted.

**L241-242:** *GLS* ***slightly outperforms the other mixing schemes*** *for some tracers, with scores above 94% for $NO_3$, $PO_4$, and $O_2$, and a mean SS of 93.16% across all tracers.*

**L255-256:** *Overall, the analysis reveals that all three mixing schemes (GLS, MY25, and LMD)* ***yield comparable and robust results*** *for key biogeochemical tracers,* ***with GLS performing slightly better in some metrics, followed closely by MY25 and LMD.***

- Line 267: The word "consistently" has been removed, and we now describe the temporal variability in performance without asserting a clear hierarchy.

**L280-281:** ***For $NH_4$, the GLS scheme tends to perform slightly better, with higher SS values across most DT configurations (Figs. S1, S2, S3). For other tracers, differences among the schemes are minimal, and no consistent ranking is evident.***

- Line 280: We revised the text to state that "all three schemes perform similarly," and avoid implying superiority of GLS.

**L293-295:** *Among the mixing schemes, GLS* ***shows somewhat lower errors*** *for $NH_4$****, specially near the coast (Fig. 3A)****, but overall all three schemes display similar behaviour, particularly in offshore regions where errors approach zero.***

- Lines 338–339: We adjusted the wording to say that discrepancies are "generally small across all schemes" without attributing lower errors to GLS specifically.

**L351-352:** *Discrepancies are **generally small across all three schemes, with GLS showing slightly smaller differences in some regions** (Fig. 6A).*

- Line 397 & 402: We removed subjective claims regarding variability or deviations in LMD and GLS, and now simply note that the schemes show similar behavior in Fig. S5.

**L409-412:** *Looking at the other mixing schemes in terms seasonal variations, the winter (December-January-February; DJF) and fall (September-October-November; SON) seasons exhibit minimal variation, **but the three mixing schemes produce largely similar patterns** (Fig. S5). **While LMD appears to show slightly more variability in fall, these differences are modest** (Fig. S5).*

**L415-416:** *Among the three schemes, **GLS tends to yield slightly lower deviations in some locations, although all remain within a comparable range.***

- Line 411: We revised this statement to acknowledge that all three schemes produce similar offline results relative to the online reference, with no single scheme being most effective across all metrics.

**L425-428: *No scheme in particular outperforms the others. For instance, the GLS scheme exhibits a slight overestimation during the fall at Point 1 and in winter at Point 2 (Fig. 9A). Meanwhile, MY25 and LMD demonstrate comparable deviations throughout the year. Overall, the results indicate that all schemes effectively reproduce vertical CHL patterns, with only minor and spatially variable differences.***

These changes are intended to better reflect the subtlety of the results and avoid overstating the significance of differences that may not be so evident.

(2) Thyng et al. 2021 mentions: "...Important factors for maintaining high offline accuracy are outputting from the online simulation often enough to resolve the advection timescale...".

In the present study, the authors selected 3-hourly output frequency for the "online" simulation, but there is no discussion on how this particular choice compares with the advection timescale. Shouldn't this be at least briefly addressed, considering how important it was in the study of Thyng et al.?

**RESPONSE:** Certainly, a brief discussion of the output frequency in relation to the advection timescale adds useful context and strengthens the methodological justification. In response, we have added a paragraph in the Methods section:

**L151-155: *Following the recommendation of Thyng et al. (2021), a 3-hourly hydrodynamic input frequency was selected to run the offline simulation. In their Gulf of Mexico simulations, these authors estimated an advection timescale of***

*approximately 5.6 hours, based on a characteristic velocity of 0.5 m/s and length scale of ~10 km. Our choice of a 3-hour interval thus falls well below this threshold, providing at least one output every ~0.5 advection timescales (Thyng et al., 2021). This frequency is therefore adequate to resolve key physical changes and ensure accurate offline interpolation of biogeochemical tracers.*

Also in the Discussion section:

**L600-613:** *A relevant consideration when using offline biogeochemical models is the temporal resolution of the hydrodynamic archive used to force tracer transport (Thyng et al, 2021). In our setup, all offline simulations were forced with 3-hourly velocity, temperature, salinity, and other physical fields extracted from the online model. This relatively fine sampling helps retaining much of the variability related to tracer advection and diffusion. Since all simulations use the same archived fields, the differences observed between the DTs cases are not due to changes in the input currents themselves, which are identical, but rather to the internal numerical treatment of tracer advection within the offline model.*

*Nevertheless, it should be noted that the hydrodynamic sampling rate used to force the offline model may limit the ability of Offline Fennel to accurately reproduce biogeochemical tracer fields. While this was not a dominant source of error in our study area, the effect could become more pronounced in highly physically variable regions, such as strongly tidal or wind-driven coastal environments. In such cases, the loss of high-frequency momentum variability between archive snapshots could degrade the fidelity of biogeochemical tracer simulations. This highlights the importance of choosing an appropriate archive sampling rate when configuring offline models, particularly for regions with energetic sub-daily dynamics.*

This addition clarifies our reasoning and explicitly connects our choice to the guidance from Thyng et al. (2021), in addition of discussing the choice.

(3) Lines 201-203: "Weights were applied to the RMSE calculation across s_rho, thus ensuring that deeper layers, which generally show less variability, did not disproportionately influence the overall calculation error"

This sounds quite arbitrary. The previous metric (SS, Eq.1) also included the vertical dimension, but somehow it didn't necessitate an arbitrary "weighing" of the vertical levels. Why is it necessary in the case of the RMSE, but not for SS?

**RESPONSE:** Our original wording may have inadvertently implied that non-uniform weighting was applied **across** the vertical levels ($s\_rho$), which is not the case. This was a miscommunication in how we described RMSE.

Thus, we now clarify that equal weights were used across the vertical levels in the RMSE calculation, and we have revised the text accordingly. Regarding the equation, the variable *"n"* represents the total number of vertical levels. This has now been corrected and explicitly stated in the manuscript here:

**L214-222:** *In this study, RMSE was computed across the vertical coordinate dimension ('s_rho'), representing depth levels. **Equal weights were used across all vertical layers, meaning that each depth level contributed equally to the overall error. This approach avoids bias toward any specific layer and provides a balanced assessment of** model performance across the entire water column.*

$$RMSE = \sqrt{\frac{1}{n}\sum_{i=1}^{n}(C_{on} - C_{off})^2} \quad \text{(Eq. 2)}$$

*where $C_{off}$ and $C_{on}$ are the time-averaged concentrations of a tracer on the 3D grid for the offline and online simulations, respectively**, and n is the total number of vertical levels.***

A few thoughts to consider:

a) It's not clear to me that the RMSE metric is needed. You already show SS, and you already show time-averaged differences (e.g. Fig.6). I personally found Fig.6 to be more informative than Fig.3. Maybe you would want to drop the RMSE?

b) If you think that the RMSE is absolutely necessary to your study, then you should list what those weights are (for each vertical level), and explain how you determined them. If you don't tell us what weights were used, then nobody can reproduce your RMSE calculation!

**RESPONSE:** Our initial motivation for including RMSE was to provide a standard and well-known scalar metrics that complements the SS by quantifying the absolute magnitude of discrepancies across the 3D domain, with a specific focus on vertical structure.

While we acknowledge that RMSE and SS partially offer overlapping information, they also serve distinct purposes. The SS metrics is normalized and reflects both spatial and vertical variability over time, making it especially effective for evaluating model skill across simulations. RMSE, in contrast, provides a direct measure of absolute error and was computed independently at each vertical level (*s_rho*), which allows us to better assess how performance varies with depth across lat-lon points of the model domain.

As presented in the manuscript, SS results are summarized in Table 1 and visualized as time series in Fig. 2, while RMSE is used to generate spatial maps in Fig. 3. We believe this combination provides a more comprehensive assessment of model performance across different dimensions. That said, we are open to simplifying or consolidating metrics if the editor recommends doing so.

In response to point (b), we have clarified in the revised text that equal weights were applied across vertical levels in the RMSE calculation, as addressed in Point (3).

(4) Lines 439-451: This passage discusses Figure 11 (that shows temporal variations in how similar the x1,x3,... cases are). The authors arbitrarily divide the 1 year timeseries into 4 "periods" associated with more/less similarity between the different cases, with

interpretations such as "this is a spin-up period" and "this is an artefact period". But in reality, changes between the 4 periods may be entirely dictated by the external forcings (rather than being a "spin-up", or a "artefact"). My recommendation would be to either (a) clarify in the text that the "4 periods" represent one possible interpretation of Fig.11 among other equally valid interpretations, or (b) conduct additional analyses to better understand these changes occurring over the year. For example, running the period Nov.2017-Oct.2018 twice (back to back) could possibly help understand whether the "higher variability" of the "blue period" (line 441) is truly a spin-up, or whether it is entirely dictated by the external forcing.

**RESPONSE:** Certainly, our interpretation of the four periods in Fig. 11 may appear overly definitive, especially given the complexity of the physical drivers involved. Our aim was offering a heuristic framework to guide interpretation, but we recognize that these divisions are not the only valid explanation and could indeed be influenced by external forcing alone.

In response, we have revised the text in Lines **476-479** (Results section) to emphasize that the segmentation into four periods is interpretive. The revised passage now includes the following clarification:

**L476-479: *We note that the four periods highlighted in Fig. 11 and Fig. S7 represent one possible interpretation of the observed temporal variability, based on apparent shifts in similarity across cases. These divisions may reflect transient behaviours such as spin-up dynamics or offline initialization effects. However, they could also be shaped by seasonal variability in the external forcings. Further investigation would be required to disentangle these factors*.**

Regarding the reviewer's suggestion to re-run the model across two consecutive years to isolate potential spin-up effects, we agree that this would be an excellent way to test the hypothesis more rigorously. However, due to time and computational constraints, we are unable to include this extended experiment in the current revision. That said, we have added a note in the Discussion section identifying this as a valuable direction for future work:

**L576-578: *In addition, a potential avenue for future research would be to repeat the one-year simulation in back-to-back cycles to better determine whether early-period variability reflects model spin-up effects or is fully driven by external forcing. Such an approach would help isolating transient initialization behaviours from recurring seasonal signals.***

(5) The study considers x1,x3,x5,x10,x15 multiples of the online time-step (line 156). The authors mention that 15x is not usable (line 160) so I'll ignore that one. They also demonstrate that the fidelity of the results is similar across x1,x3,x5,x10 (Table 1). So x10 is the option that most people would choose (it is just as good based on Table 1, and it gives the highest computational gain). But somehow, the authors chose to show results from 5x in most of their figures: Figs.2,3,5,6,8,9. Why not show the x10 case in those

figures? Ultimately, 10x is what a user of "Offline Fennel" would pick, so that's what they would want to see.

**RESPONSE:** It is true that the x10 configuration delivers the greatest computational efficiency while maintaining strong fidelity to the online reference, as shown in Table 1. However, our choice to present the x5 configuration in most of the figures (Figs. 2, 3, 5, 6, 8, and 9) was based on its conservative yet efficient balance. The x5 case offers a substantial computational gain (approximately a 75% reduction in simulation time) while minimizing any risk of temporal resolution artifacts that may arise from further increasing the time step.

This choice is especially relevant because the present study aims to propose a setup that can be applied in a variety of regional configurations, including areas with more energetic dynamics. In such cases, increasing the time step (e.g., using x10) may lead to errors in representing processes that require higher temporal resolution - for instance, during strong advection events or under intense atmospheric forcing.

Moreover, the additional time savings between x5 and x10 is relatively modest in absolute terms, typically decreasing simulation time from around 50 minutes to 30 minutes. Given this diminishing return, we believe x5 represents a practical and reliable default option for users who are new to the Offline Fennel model or who prefer a cautious first implementation.

Offline Fennel users may have different priorities: some may seek maximum speed-up (favoring x10), while others may prefer a more conservative implementation (favoring x5). By selecting x5 for the main figures, we aimed to present results from a robust yet efficient case that we believe many first-time users would find reassuring. That said, we agree that x10 is a very reasonable and attractive choice for many applications, and we provide comparative results for the x10 configuration in other figures so that users can evaluate tradeoffs and decide based on their specific needs.

OTHER COMMENTS

(6) Line 1: Aren't "high-performance" and "computationally efficient" synonyms?

Do we really need to see both inside the title?

**RESPONSE:** While the terms "high-performance" and "computationally efficient" can sometimes be used interchangeably, in this context we intended them to highlight two distinct aspects of the Offline Fennel model:

- High-performance refers to the scientific performance of the model, i.e., its ability to reproduce the online results with high fidelity.
- Computationally efficient refers to its low computational cost and speedup relative to the online configuration.

Actually, the two terms were intended to capture two critical but different aspects: accuracy vs. speed. We believe both characteristics are relevant and complementary.

(7) Line 15: The authors take for granted that the reader is already familiar with the concept of an "offline" model.

Instead, how about briefly explaining what "offline" refers to?

E.g.: "...present the "offline" Fennel model, a biogeochemical model relying on previously-calculated hydrodynamical outputs from the Regional Ocean..."

**RESPONSE:** Certainly, the term "offline model" may not be immediately clear to all readers, particularly those less familiar with ocean modeling workflows.

In response, we have revised the sentence on Line 15 (Abstract section) to include a concise explanation of the term:

**L15:** *Here, we present the Offline Fennel model, **a biogeochemical model that relies on previously computed physical** fields from the Regional Ocean Modeling System (ROMS).*

This change improves clarity for a broader readership and aligns closely with the reviewer's helpful suggestion.

(8) Line 52: Grobe et al. 2020 is cited in the sentence but absent from the "References".

**RESPONSE:** We apologize. We have corrected the References list and added the missing Große et al. (2019-2020) references.

**L823-827:** *Große, F., Fennel, K., and Laurent, A.: Quantifying the relative importance of riverine and open-ocean nitrogen sources for hypoxia formation in the northern Gulf of Mexico. Journal of Geophysical Research: Oceans, 124 (8), 5451–5467. https://doi.org/10.1029/2019JC015230, 2019.*

*Große, F., Fennel, K., Zhang, H., and Laurent, A.: Quantifying the contributions of riverine vs. oceanic nitrogen to hypoxia in the East China Sea. Biogeosciences, 17 (10), 2701–2714. https://doi.org/10.5194/bg-17-2701-2020, 2020.*

(9) Line 73: "...that utilizes advanced physical and numerical algorithms (Shchepetkin, 2003)"

"Advanced" is a subjective word, and I'm not sure it is appropriate to use it when citing a paper that was published 22 years ago.

**RESPONSE:** The term "advanced", which may indeed seem subjective, especially in the context of a paper that is over two decades old. To address this concern, we have revised the sentence to be more neutral and objective:

**L74:** *...that utilizes* **well-established** *physical and numerical algorithms (Shchepetkin, 2003)...*

This revision maintains the intended meaning but removes the potentially outdated or subjective term "advanced", making the statement more appropriate for the context.

(10) Lines 86-87: "Additionally, improvements were made to fix the handling of climatology files for the bottom-depth layer to ensure more accurate simulations."

This sentence is very vague; I don't know what it is referring to. I would recommend removing the sentence, or if it is truly important, then clarify what it means.

**RESPONSE:** The previous version of Thyng et al. (2021) model introduced artifacts in the bottom layer due to the way climatology files were handled. Specifically, a time-shifting error occurred when the model accessed climatology data for sea surface height and 3D momentum, which affected the bottom layers and propagated a bias towards the surface, distorting biogeochemical tracer concentrations. This is addressed in Appendix A, but for clarity, we have revised this sentence explaining this also in the Methods section:

**L89-93:** *In the previous model version by Thyng et al. (2021), a shift occurred when processing climatology fields, leading to a bias that propagated from the bottom toward the surface, affecting tracer concentrations. The offline model was modified to prevent the simulation from accessing subsequent time step values for sea surface height and 3D momentum climatologies, thereby eliminating this unintended artifact.*

We believe this clarification provides the necessary context for readers to understand the change without need to read the Appendix.

(11) Line 140: "The physical forcing conditions for the offline..."

"Physical" is a very broad word. Did you mean something more specific, like "atmospheric", or perhaps "surface"?

**RESPONSE:** We have revised the sentence and use now the term "surface" to be more precise:

**L145-147:** *The physical* **surface** *forcing conditions for the offline biogeochemical model included solar shortwave radiation flux, surface net heat flux, surface u-momentum stress, and surface v-momentum stress*, **which were derived from the corresponding online simulation outputs**.

This revision specifies the types of forcing conditions used in the offline model and ensures clearer communication of the offline model setup.

(12) Line 154: "...the LMD simulations excluded these additional forcing fields..."

Don't you want to briefly explain why LMD is treated differently from GLS and MY25 in your study?

I assume it is because LMD wasn't included in Thyng et al.'s study?

**RESPONSE:** The reason for treating the LMD simulations differently is indeed to facilitate a direct comparison with GLS and MY25, while also assessing the impact of additional forcing fields on model behaviour. This decision allowed us to evaluate the effects of their absence in the model outputs and compare the results under two different forcing scenarios —one with the additional fields (GLS and MY25) and one without (LMD).

We have revised the sentence to reflect this clarification and provide the necessary context for the different treatment of the simulations:

**L16-168:** *In contrast, the LMD simulations excluded these additional forcing fields, **which allowed us to evaluate** the effects of their absence **and compare** the model outputs under different forcing scenarios. This treatment also ensured a consistent comparison with the setup used by Thyng et al. (2021) for the GLS and MY25 simulations.*

(13) Line 274: "O2 systematically shows the lowest errors..."

How can we say that O2 has "lower errors" than the other variables?

**RESPONSE:** When we state that $O_2$ shows lower errors when describing results, here referring to Fig. 3, we are comparing to the magnitude of the differences relative to the typical climatological values observed in the simulation. These concentrations generally range between 200 and 400 $mmol \cdot m^{-3}$, as shown in Fig. 5, and average around 200-300 $mmol \cdot m^{-3}$ throughout the water column profiles, as seen in Fig. 7. The maximum differences in $O_2$ across simulations are typically below 10 $mmol \cdot m^{-3}$, which represents only a small fraction of the natural variability of the tracer.

Moreover, these differences are primarily restricted to the coastal region, whereas offshore areas show minimal and spatially consistent discrepancies. This localized pattern, combined with the relatively small deviations within the expected concentration range and variability, supports our conclusion that $O_2$ exhibits lower model differences compared to other tracers.

(14) Line 297: Is "accuracy" the most appropriate word in this context? Figure 4 is evaluating the degree of similarity among the different simulations, while "accuracy" typically means something else.

**RESPONSE:** Certainly, "accuracy" typically refers to how close a model's predictions are to the true or observed values, whereas Figure 4 is assessing the degree of similarity among different simulations, which is more about comparing the outputs rather than evaluating their absolute correctness.

Therefore, we have changed the word "accuracy" by "similarity" to better reflect the content of the figure:

**L309-310:** *The figure illustrates that simulations using the same mixing scheme (GLS, LMD, or MY25) exhibit the highest **similarity**,…*

We hope this clarification resolves the issue, and we appreciate the reviewer's attention to precision in terminology.

(15) Lines 321-322: "The main plots presented here are from the surface layer, as it is the most impacted by forcing, given that the offline simulations are forced with the physical outputs from the online model. This means that the surface layer is where most bias can be introduced."

I don't understand this passage. What "forcing" are you referring to, exactly? The wind stress? Do you mean that the surface is the layer having the strongest temporal variability, and therefore it is harder for the offline model to perfectly capture all this temporal variability?

**RESPONSE:** The term "forcing" here refers to the physical surface conditions used to drive the offline simulation, specifically shortwave solar radiation, surface net heat flux, and u- and v-momentum, all of which derive from the online model output.

It is true that the surface layer experiences the strongest temporal and spatial variability, especially due to the above-mentioned drivers and also to atmospheric forcings, which makes it particularly challenging for the offline model to replicate the online reference accurately. This is why we focus our main analysis on the surface layer. For completeness, we also include complementary figures for the bottom layer and vertical profiles, allowing assessment of model behaviour throughout the water column.

We have revised the passage for clarity:

**L335–337:** *The main plots **here presented focus on** the surface layer, **which** is the most **dynamically variable region due to exposure to the physical model forcing (e.g., shortwave solar radiation, net heat flux, u- and v- momentum). This variability increases the likelihood of discrepancies between the online and offline simulations.***

(16) Figure 10: Given that the x15 case is not viable and that healthy results could not be obtained from it (line 431), are you sure it's a good idea to include it in Figure 10? It's a bit misleading to provide values for a case that doesn't work. (Same comment for Table S1.)

**RESPONSE:** It is true that including the x15 case in Fig. 10 may be misleading, particularly since this configuration did not write results, as noted in line 431. Our original intent was to illustrate the full range of simulation attempts. However, we understand that displaying results for an unviable case could cause confusion.

In response, we have removed the x15 case from Fig. 10 to ensure clarity. However, we have retained it in Table S1 (supplementary materials) for ease of completeness, as it documents the full set of configurations explored and highlights the performance limits of the offline model. This distinction helps clarifying which configurations are viable for analysis and which are not.

**L172-174 (Methods):** *A DT 15 times longer than the online time-step led to unstable writing of solutions.* ***As such, while this case was initially tested, it was excluded from the analysis figures to avoid misleading interpretations.***

(17) Line 550: "A particularly intriguing aspect of the results is the lack of growing error over time..."

The simulations were limited to one year (line 101) and this would contribute to small errors. The fact that the model domain is relatively small (with one side being a coastline with prescribed riverine fluxes) probably also helps, right?

**RESPONSE:** The relatively short duration of the simulations (one year), the limited spatial extent of the model domain, and the presence of a coastline with prescribed riverine fluxes may indeed help constrain variability and reduce the potential for error growth over time. These factors likely contribute to the observed stability in the offline simulations.

That said, we also view the absence of growing error as a positive indicator of model performance under the given setup. This result suggests that biological variability is effectively constrained by the physical conditions imposed by the online model, and that the offline model's simplification of physical–biological feedbacks does not result in the progressive accumulation of errors.

We have updated the paragraph to reflect this:

**L580-586:** *A particularly intriguing aspect of the results is the lack of growing error over time in the Offline Fennel model simulations, as typically observed in other studies, like data assimilation schemes (Berry and Harlim, 2017).* ***This may partly reflect the relatively short simulation period (one year), the model domain, and the stabilizing influence of boundary conditions such as prescribed riverine fluxes. However, it also suggests that biological variability remains well constrained by the physical state and that the simplification of feedback mechanisms in the offline framework does not result in significant error accumulation (Béal et al., 2010). Further investigation is needed to explore this phenomenon more thoroughly and assess whether such a stability persists across different configurations and longer timeframes.***

---

## Referee Report (RR1)

Review of *Offline Fennel: A High-Performance and Computationally Efficient Biogeochemical Model within the Regional Ocean Modeling System (ROMS)* by J. Crespín, J. Solé and M. Canals

This manuscript describes an offline implementation of a well-stablished biogeochemical model, the Fennel model, and offers an evaluation of its performance against the same biogeochemical model coupled online to an ocean model configured for the Gulf of Mexico, from which physical variables are extracted to run the offline case. The offline model performance is evaluated considering different time steps. In addition, the role of three vertical mixing scheme closures is explored.

This offline model is a valuable tool that reduces the computational cost of running biogeochemical models. It is also appreciated that it is publicly available to the community. After evaluating the revised manuscript version as well as the replies to initial reviewers, I find it well-structured, relevant and fits well within the scope of the journal, for which I recommend it for publication. I have, however, a few comments that aim to enhance the clarity of the manuscript.

(Line numbering in comments below refers to the latest version of May 17 with tracked changes)

MAJOR COMMENTS

i)  I am confused about how biogeochemical tracer diffusion is implemented in the offline model regarding the three points below, which should only require some clarifications in the text.

- Authors mention the usage of three different mixing schemes, but these are vertical mixing schemes while nothing is said about horizontal mixing at subgrid scale. Please add a line or two about it.
- In L126 authors refer to the GLS scheme but this is a generic scheme. Authors should specify if it was implemented as k-kl (a specific case of which is the MY25 scheme), as k-ε or k-ω as indicated in Warner et al. (2005), as well as the parameters used (or refer to a publication that uses the same parameters) for reproducibility of results.
- L161-L168. Authors mention that AKs, AKt, and AKv are used to force the offline model in GLS and MY25 mixing schemes but not the LMD one. I find this very confusing. The aim of using a vertical mixing scheme is to obtain a vertical diffusion coefficient to use for subgrid-scale vertical mixing in the biogeochemical tracer equations (e.g., eq. 9 in Fennel et al. 2022). Which of the three (AKs, AKt, and AKv) is being used to diffuse vertically biogeochemical tracers in GLS and MY25 mixing schemes? Why do you need the TKE and the generic length scale output in the offline model if the vertical diffusivity is already provided? Additionally, the text reads as no vertical diffusivity is used in the LMD scheme case, is this correct?

ii) Reviewer 2 had a concern about the additional value of showing the RMSE besides the SS. The concern arises because the SS is the RMSE normalized by the online values, therefore both properties quantify differences in amplitude between the online and offline. Authors could replace Table 1 by a Taylor diagram (Taylor 2001), offering a visual and faster comparison between simulations (DT and vertical mixing scheme) to the reader. This diagram shows RMSE but also the correlation coefficient, a metric on how the online and offline values covary in time and/or space, which is currently lacking in the evaluation. In this case, statistics for both p and RMSE should be weighted by volume given that it varies among grid cells (I believe depth levels are unevenly spaced in the model) and tracer concentrations are per unit volume. Volume-weighted averages should be also used for the computation of the SS time series in Fig. 2.

I also agree with reviewer 2 that the results shown in Fig. 3, and its related discussion (L286-L295), are applicable to Fig. 6, which also seems more relevant to me. The only differences are the discrepancies in $PO_4$ and $NO_3$ offshore, which are not very relevant to the manuscript and would likely be unveiled in Fig. 6 if considering the full water column.

Finally, related to model evaluation of results shown in Fig. 4, how is equation (1) applied when two offline simulations are compared? That is, which values are used in the denominator to normalize the RMSE? Could this be related to the fact that the heatmap is not exactly symmetric?

MINOR COMMENTS

L15. Comma after fields.

L20. Express the time reduction in percentage since it is more meaningful as a general message.

Authors could also mention in the abstract the fact that $NH_4$ is a challenging tracer to simulate accurately offline since its timescale of change is faster than other tracers. This is a nice general result applicable to other offline biogeochemical model implementations.

L90. Which type of shift? A time shift?

L121. Add "vertical" before "mixing".

L131-L132. Instead of listing all variables, authors could say that the Fennel model variables are included except those involved in carbon pools (which imply using ALK). This is a more direct way for the reader to understand what is exactly included and avoids redundancy since all Fennel model variables are listed in L100-102.

L147. Climatology over which time period?

L147. Explain what zeta, ubar- and vbar- velocity properties are.

L174-L175. Delete last sentence to avoid too much redundancy. The value of the barotropic time step is already stated in L107.

L124. Delete "subgrid-scale turbulent mixing closure scheme" since all three schemes are.

L309. Remove "(GLS, LMD, or MY25)" to avoid redundancy.

L330. Specify the time length of the averaged output.

Figure 8. I think panels C are redundant differences from B are shown in A. Also, in caption remove "(x5 DT)" after "offline" to avoid repetition. Finally, consider a colorblind friendly colorbar in panels B such as the cmocean ones (Thyng 2016).

REFERENCES

Fennel, K., Mattern, J.P., Doney, S.C. et al. (2022). Ocean biogeochemical modelling. *Nat Rev Methods Primers* 2, 76. https://doi.org/10.1038/s43586-022-00154-2.

Taylor, K. E. (2001). Summarizing multiple aspects of model performance in a single diagram, *J. Geophys. Res.*, 106(D7), 7183–7192, doi:10.1029/2000JD900719.

Thyng, K. M., Greene, C. A., Hetland, R. D., Zimmerle, H. M., & DiMarco, S. F. (2016). True colors of oceanography. *Oceanography*, 29(3), 10.

Warner, J.C. et al. (2005). Performance of four turbulence closure models implemented using a generic length scale method, *Ocean Modelling*, Volume 8, Issues 1–2, 2005, Pages 81-113.

---

## Author Response (AR2)

**RESPONSE TO REVIEWER #3**

This manuscript describes an offline implementation of a well-stablished biogeochemical model, the Fennel model, and offers an evaluation of its performance against the same biogeochemical model coupled online to an ocean model configured for the Gulf of Mexico, from which physical variables are extracted to run the offline case. The offline model performance is evaluated considering different time steps. In addition, the role of three vertical mixing scheme closures is explored.

This offline model is a valuable tool that reduces the computational cost of running biogeochemical models. It is also appreciated that it is publicly available to the community. After evaluating the revised manuscript version as well as the replies to initial reviewers, I find it well-structured, relevant and fits well within the scope of the journal, for which I recommend it for publication. I have, however, a few comments that aim to enhance the clarity of the manuscript.

(Line numbering in comments below refers to the latest version of May 17 with tracked changes)

**RESPONSE:** We would like to thank the reviewer for the constructive and detailed comments, which have helped us improve the clarity and robustness of the manuscript. Below we address each of the points raised and indicate the corresponding changes made in the manuscript. Line numbering in refers to the updated revised version with tracked changes from 30[th] June.

**MAJOR COMMENTS**
i) I am confused about how biogeochemical tracer diffusion is implemented in the offline model regarding the three points below, which should only require some clarifications in the text.

• Authors mention the usage of three different mixing schemes, but these are vertical mixing schemes while nothing is said about horizontal mixing at subgrid scale. Please add a line or two about it.

**RESPONSE:** In our setup, harmonic horizontal mixing of velocities and tracers is applied along geopotential surfaces using the 'UV_VIS2', 'TS_DIF2', 'MIX_GEO_UV', and 'MIX_GEO_TS' options. We also employ the 'TS_MPDATA' advection scheme – recommended by Thyng et al. (2021) – to minimize numerical diffusion. The 'DIFF_GRID' and 'VISC_GRID' options enable grid-dependent variations in lateral diffusivity and viscosity. All simulations use this configuration to ensure full consistency and comparison between experiments.

We have included these lines for clarity in the Methods section.

**L132-135:** *Harmonic horizontal mixing of velocities and tracers was applied along geopotential surfaces, together with the TS_MPDATA advection scheme to minimize numerical diffusion (Thyng et al., 2021). Grid-dependent diffusivity and viscosity were also enabled via DIFF_GRID and VISC_GRID. All*

*configuration files, including exact parameters for each mixing scheme, are available for full reproducibility (Crespin, 2025b).*

All CPP options for each simulation are publicly archived on Zenodo (DOI: 10.5281/zenodo.14930137).

• In L126 authors refer to the GLS scheme but this is a generic scheme. Authors should specify if it was implemented as k-kl (a specific case of which is the MY25 scheme), as k-e or k-w as indicated in Warner et al. (2005), as well as the parameters used (or refer to a publication that uses the same parameters) for reproducibility of results.

**RESPONSE:** We implement the GLS scheme in its k–ε configuration, following Warner et al. (2005). The key exponent parameters are set to GLS_P = 3.0, GLS_M = 1.5, and GLS_N = −1.0. All configuration files for online and offline simulations (including exact parameters for each field), are openly available on the Zenodo repository (DOI: 10.5281/zenodo.14930137; https://zenodo.org/records/14930138), as well as input files, allowing full reproducibility of the study.

We have specified this in the manuscript as follows:

**L129-130:** *The GLS scheme in our simulations corresponds to the k–ε configuration (Warner et al., 2005), defined by the exponent values 'GLS_P' = 3.0, 'GLS_M' = 1.5, and 'GLS_N'= -1.0.*

• L161-L168. Authors mention that AKs, AKt, and AKv are used to force the offline model in GLS and MY25 mixing schemes but not the LMD one. I find this very confusing. The aim of using a vertical mixing scheme is to obtain a vertical diffusion coefficient to use for subgrid-scale vertical mixing in the biogeochemical tracer equations (e.g., eq. 9 in Fennel et al. 2022). Which of the three (AKs, AKt, and AKv) is being used to diffuse vertically biogeochemical tracers in GLS and MY25 mixing schemes? Why do you need the TKE and the generic length scale output in the offline model if the vertical diffusivity is already provided? Additionally, the text reads as no vertical diffusivity is used in the LMD scheme case, is this correct?

**RESPONSE:** We thank the reviewer for this important question and apologize for the confusion. All details on the use of the AKXCLIMATOLOGY and MIXCLIMATOLOGY flags are documented in Thyng et al. (2021). Below we clarify their behavior in the offline implementation:

With these flags enabled (GLS & MY25 offline simulations):
- The offline model reads the pre-computed fields (AKs, AKt, AKv, TKE, and GLS) directly from the online parent simulation at 3-h intervals.
- Of these, AKt (vertical temperature diffusivity) is used for subgrid-scale vertical mixing of all passive tracers, including biogeochemical variables. AKs (vertical salinity diffusivity) and AKv (vertical viscosity) influence only salinity and momentum, respectively, and do not modify passive tracer

advection (Thyng et al., 2021). TKE and GLS are imported for diagnostic consistency.

Without climatology flags (LMD):
- The offline model recomputes its own vertical mixing coefficients (AKs, AKt, AKv) internally from the LMD closure, using the exact same parameter values as in the online run.
- Thus, vertical diffusivity remains active, computed on-the-fly rather than read in.

All three configurations therefore supply the necessary vertical diffusivity for biogeochemical tracers: via climatology read-in for GLS/MY25, and via internal computation for LMD. We retain AKt, AKs, AKv, TKE, and gls in the GLS/MY25 cases solely to adhere to the original offline framework (Thyng et al., 2021) and for diagnostics. In Thyng et al. (2021) they report that only AKs had a minor impact on tracer accuracy, while Akt, Akv, TKE, and GLS did not.

We have updated the manuscript for clarity:

**L171-182:** *For the GLS and MY25 simulations, the offline model was forced by additional vertical mixing parameters – namely vertical salinity diffusion (AKs), temperature vertical diffusion coefficient (AKt), and vertical viscosity coefficient (AKv), –, all of which influence sub-grid-scale vertical mixing. These fields are obtained from the online parent run via the 'AKXCLIMATOLOGY' CPP flag, which ingests the 3-h climatologies of AKs, AKt, and AKv. In addition, the 'MIXCLIMATOLOGY' flag is used to import the generic length scale (GLS) and turbulent kinetic energy (TKE) coefficients from the online simulation.*

*In contrast, LMD simulations omit these climatology flags, so the offline model recomputes its own AKs, AKt, and AKv internally by using the same turbulence-closure parameters defined in the online configuration. This ensures that vertical diffusivity remains active under all schemes while enabling a direct test of sensitivity to externally prescribed versus internally computed mixing fields. This treatment also mirrors the approach of Thyng et al. (2021) for the GLS and MY25 cases; for full implementation details of these flags, refer to Thyng et al. (2021).*

ii) Reviewer 2 had a concern about the additional value of showing the RMSE besides the SS. The concern arises because the SS is the RMSE normalized by the online values, therefore both properties quantify differences in amplitude between the online and offline. Authors could replace Table 1 by a Taylor diagram (Taylor 2001), offering a visual and faster comparison between simulations (DT and vertical mixing scheme) to the reader. This diagram shows RMSE but also the correlation coefficient, a metric on how the online and offline values covary in time and/or space, which is currently lacking in the evaluation. In this case, statistics for both p and RMSE should be weighted by volume given that it varies among grid cells (I believe depth levels are unevenly spaced in the model) and tracer concentrations are per unit volume. Volume-weighted averages should be also used for the computation of the SS time series in Fig. 2.

I also agree with reviewer 2 that the results shown in Fig. 3, and its related discussion (L286-L295), are applicable to Fig. 6, which also seems more relevant to me. The only differences are the discrepancies in $PO_4$ and $NO_3$ offshore, which are not very relevant to the manuscript and would likely be unveiled in Fig. 6 if considering the full water column.

Finally, related to model evaluation of results shown in Fig. 4, how is equation (1) applied when two offline simulations are compared? That is, which values are used in the denominator to normalize the RMSE? Could this be related to the fact that the heatmap is not exactly symmetric?

**RESPONSE:** We appreciate the reviewer's insightful suggestions regarding the evaluation metrics used in our study. In response to each of the reviewer's points, we have made the following enhancements:

1. **Volume-weighted metrics:** We acknowledge the importance of using volume-weighted averages for both RMSE and SS, especially given the variability in grid cell depths and tracer concentrations. We have implemented these changes in our calculations, ensuring that all skill metrics are now computed using volume-weighted averages. This adjustment is detailed in the Methods section (Subsection 2.5, Lines 212-242), where we explain the weighting procedure applied to each metric as follows:

   **L213-242:** *We evaluate the Offline Fennel model using two complementary, volume-weighted diagnostics: a skill score (SS) and the root-mean-square error (RMSE). Both metrics account for the true physical volume ($V_{i,j,k}$) of each grid cell, thereby avoiding biases due to varying horizontal areas or layer thicknesses.*

   *Each cell volume ($V_{i,j,k}$) is computed as the product of the horizontal ROMS grid spacings ($\Delta x_i$, $\Delta y_j$) and the vertical thickness ($\Delta z_{i,j,k}$), which is calculated from the difference in model layer depths ($z_w$) at each horizontal location (i,j).*

   *SSs are a widely used metric for evaluating model performances (Bogden et al., 1996; Hetland, 2006). To assess the performance of our Offline Fennel model, we applied the following equation (**Eq. 1**), adapted from Thyng et al. (2021):*

   $$SS = 1 - \sqrt{\frac{\sum_{i,j,k} V_{i,j,k} \times \left(C(t) - C_{ref}(t)\right)^2}{\sum_{i,j,k} V_{i,j,k} \times C_{ref}(t)^2}} \quad (Eq.\ 1)$$

   *where C(t) and $C_{ref}$(t) are the concentrations of a tracer of a tracer at time t in the compared and reference simulations, respectively – typically representing offline (uncoupled) and online (coupled) configurations. The sums are performed over all spatial and vertical dimensions, and results are volume-weighted. This yields a time series of SS values that tracks the temporal evolution of model performance. A time-mean SS can be computed*

*by averaging over the simulation period, providing a single, scalar measure of overall model accuracy.*

*To complement the SS analysis, RMSE was equally employed as a metric to assess the accuracy of offline simulations compared to online results (**Eq. 2**). RMSE provides insight into the magnitude of errors by measuring the square root of the average squared differences between offline and online simulation results.*

$$RMSE = \sqrt{\frac{\sum_{i,j,k} V_{i,j,k} \times \left(C - C_{ref}\right)^2}{\sum_{i,j,k} V_{i,j,k}}}$$

*(Eq. 2)*

*where C and $C_{ref}$ are the time-averaged concentrations of a tracer on the 3D grid for the offline and online simulations, respectively. Because each grid cell's contribution is proportional to its volume, this RMSE reflects the true three-dimensional error structure.*

*Together, these volume-weighted SS and RMSE metrics provide a robust evaluation of model performance, capturing both relative and absolute discrepancies while avoiding biases caused by unequal grid cell sizes.*

Moreover, the figures based on these updated metrics have been revised accordingly: Figs. 2, 3, and 4, as well as Supplementary Figs. S1, S2, and S3, and Table 1.

While the overall results remain consistent with the previous version, we have adjusted some numerical values in the following paragraphs of the Results section to reflect the new metrics. The changes are highlighted in **bold** for clarity:

**L263-286:** *Table 1 summarizes the mean SSs computed using Eq. 1 for key biogeochemical tracers. Across all mixing schemes, the simulations demonstrate high accuracy, with minimal differences between configurations. GLS slightly outperforms the others in some tracers, with scores **above 95% for $NO_3$, $PO_4$, and $O_2$, and a mean SS of 92.92% across all tracers.** CHL scores hover around **91%,** highlighting the scheme's ability to capture primary production dynamics effectively. However, $NH_4$ exhibits lower SSs, ranging from **83.21% to 83.53%.***

*The LMD scheme similarly produces robust results, particularly for **$O_2$, $PO_4$, and $NO_3$ with SSs of 98.68%, 96.00% and 95.44%, respectively**. Its mean SS is slightly lower (**92.79%),** reflecting solid tracer performance overall, although $NH_4$ again shows the weakest accuracy, with SSs between **82.53% and 82.94%**.*

*Results for the MY25 scheme align closely with those of GLS and LMD, yielding SSs of **96.12%** for $NO_3$ and **96.86%** for $PO_4$. The mean SS for MY25 is **92.86%,** underscoring its comparable performance. While $NH_4$ again exhibits lower SSs, the values still represent good model performance, as they remain above **80%.***

**L332-335:** *In relation to typical tracer concentrations in the study area, $O_2$ systematically shows the lowest errors across the domain, with only small and localized increases near the coast. […]*

**L337-338:** *Finally, $NH_4$ presents the highest error levels **when considering typical concentrations in the region,** which range from 0 to 5 $mmol·m^{-3}$ at the surface and can reach up to 20 $mmol·m^{-3}$ at depth. […]*

**L344-345:** *In contrast, comparisons across different mixing schemes show decreased SSs, dropping **to 92%** in some cases between GLS and LMD simulations.*

2.  **Addition of a Taylor diagram:** we concur that a Taylor diagram offers a comprehensive visual representation of model performance by integrating RMSE, correlation coefficients, and standard deviation metrics. To enhance our analysis, we have added this diagram to complement Table 1, providing a clearer comparison of the models' performance metrics across spatial and vertical dimensions.

**L261-271:** *To assess model performance, we first present a Taylor diagram (**Fig. 2**) (Taylor, 2001) that illustrates the volume-weighted and normalized statistics averaged across all biogeochemical variables. This diagram highlights a strong agreement between the offline simulations and the online parent model. The offline configurations show higher standard deviation values compared to the coupled reference (0.26 across all vertical mixing schemes), exhibiting relative standard deviations slightly exceeding 1.*

*GLS and LMD demonstrate remarkably similar performance, characterized by standard deviations ranging from 1.107 to 1.123, low centered root mean square error (RMSE) values around 0.25, and high correlation coefficients (r > 0.98). The MY25 scheme shows slightly higher RMSE values (up to 0.29) and marginally lower correlation coefficients, but these differences are minimal and do not significantly detract from its overall performance. Furthermore, differences between DTs are negligible across all cases (**Fig. 2**).*

3.  **Relevance of Fig. 3 and Fig. 6:** Following the update of the skill metrics equations to be volume-weighted (as detailed in point 1), we have chosen to retain both figures in our analysis. This decision aims to provide readers with a more comprehensive understanding of the various metrics and the differences observed across the different experiments and simulations.

4. **Comparison of offline simulations:** In relation to the application of Equation 1 for comparing two offline simulations, we would like to clarify that the denominator used for normalizing the RMSE is based on the SS reference series defined by the first simulation in the comparison. We have revised Equation 1 to incorporate this clarification, defining C and $C_{ref}$ to reduce potential confusion for readers (please see the updated equation in point 1).

**MINOR COMMENTS**

L15. Comma after fields.

**RESPONSE:** Noted and corrected.

L20. Express the time reduction in percentage since it is more meaningful as a general message. Authors could also mention in the abstract the fact that NH4 is a challenging tracer to simulate accurately offline since its timescale of change is faster than other tracers. This is a nice general result applicable to other offline biogeochemical model implementations.

**RESPONSE:** Thank you for the suggestion. We have included the time reduction in percentage and now also mention the challenging aspect about modelling NH4 as follows (changes in **bold**):

**L19-20:** *By leveraging physical hydrodynamic outputs, we ran the Offline Fennel model using various time-step multiples from the coupled configuration, significantly enhancing computational efficiency **and reducing simulation computational time by up to 87%.***

**L25-26:** ***A significant challenge identified was the simulation of ammonium ($NH_4$), which exhibited the largest discrepancies due to its rapid turnover timescale compared to other tracers.***

L90. Which type of shift? A time shift?

**RESPONSE:** Yes, thank you. We have clarified that the shift refers to time, as follows (changes in **bold**):

**L90-91**: *In the previous model version, a **time** shift occurred when processing climatology fields, leading to a bias that propagated from the bottom toward the surface, affecting tracer concentrations.*

L121. Add "vertical" before "mixing".

**RESPONSE:** Noted and added (in **bold**).

**L123:** *To evaluate the offline model performance, online simulations were conducted using three different **vertical** mixing schemes.*

L131-L132. Instead of listing all variables, authors could say that the Fennel model variables are included except those involved in carbon pools (which imply using ALK). This is a more direct way for the reader to understand what is exactly included and avoids redundancy since all Fennel model variables are listed in L100-102.

**RESPONSE:** Noted and corrected as follows (changes in **bold**).

**L137-139:** *For the biogeochemical implementation, we used the same configuration for both online and offline simulations to ensure comparability. The state variables incorporated* **all Fennel model tracers except those involved in carbon pools (e.g., alkalinity).**

L147. Climatology over which time period?

**RESPONSE:** This file is not a traditional climatology but rather a 3-hourly forcing file that is incorporated in ROMS through the climatology mechanism (Thyng et al., 2021). We have clarified the time period in the manuscript, as follows (changes in **bold**):

**L159-160:** *Following the recommendation of Thyng et al. (2021), a 3-hourly hydrodynamic input frequency was selected to run the offline simulation* **for both the physical and climatology forcing files.**

The details of each forcing file are described in the Appendices.

L147. Explain what zeta, ubar- and vbar- velocity properties are.

**RESPONSE:** Noted.

**L154-155:** *The climatology forcing incorporated variables such as* **free surface elevation (zeta), vertically integrated u-momentum component (ubar), vertically integrated v-momentum component (vbar)**, *[…]*

L174-L175. Delete last sentence to avoid too much redundancy. The value of the barotropic time step is already stated in L107.

**RESPONSE:** Noted and removed as follows:

**L183-188:** *Offline simulations were run with varying multiples of the online DT (x1, x3, x5, x10, and x15) to improve computational efficiency, until the results became unstable. Given that the baroclinic DT of the online simulation was 60 seconds, these corresponded to offline baroclinic DTs of 60 s, 180 s, 300 s, 600 s and 900 s. A DT 15 times longer than the online time-step led to unstable writing of solutions. As such, while this case was initially tested, it was excluded from analysis figures and tables to avoid misleading interpretations.*

L124. Delete "subgrid-scale turbulent mixing closure scheme" since all three schemes are.

**RESPONSE:** Noted and removed as follows:

**L124-125:** *the Large–McWilliams–Doney (LMD) mixing scheme, also known as the K-Profile Parameterization, which is a subgrid-scale turbulent mixing closure scheme (Large et al., 1994) […]*

L309. Remove "(GLS, LMD, or MY25)" to avoid redundancy.

**RESPONSE:** Noted and removed as follows:

**L343-345:** *The figure illustrates that simulations using the same mixing scheme (GLS, LMD, or MY25) exhibit the highest similarity, with SSs […]*

L330. Specify the time length of the averaged output.

**RESPONSE:** The revised sentence now specifies the time resolution clearly: (changes in **bold**):

**L371-372:** *For consistency, all plots were generated using **3-hourly** 'avg' output files from ROMS**, which were then daily averaged.***

Figure 8. I think panels C are redundant differences from B are shown in A. Also, in caption remove "(x5 DT)" after "offline" to avoid repetition. Finally, consider a colorblind friendly colorbar in panels B such as the cmocean ones (Thyng 2016).

**RESPONSE:** We acknowledge the redundancy but decided to retain both panels B and C to help readers unfamiliar with typical chlorophyll levels in the region better interpret the differences shown in panel A. As suggested, we have removed "(x5 DT)" from the caption and replaced the colormap with the colorblind-friendly 'algae' from the cmocean package (Thyng et al., 2016).

**Figure 8 caption:** *Seasonal maps of chlorophyll concentrations (CHL) for Generic Length Scale (GLS) mixing simulations using x5 time-step (DT). (A) Displays the difference in concentrations between offline (x5 DT) and online simulations, as indicated by the coolwarm color scale on the right. (B) Shows chlorophyll concentrations for the online simulation, while (C) presents concentrations for the offline simulation (x5 DT). **Panels (B) and (C) share the same color scale from the cmocean package ('algae').** Seasonal designations are as follows: DJF (December-January-February, winter), MAM (March-April-May, spring), JJA (June-July-August, summer), and SON (September-October-November, fall).*

REFERENCES

Fennel, K., Mattern, J.P., Doney, S.C. et al. (2022). Ocean biogeochemical modelling. Nat Rev
Methods Primers 2, 76. https://doi.org/10.1038/s43586-022-00154-2.
Taylor, K. E. (2001). Summarizing multiple aspects of model performance in a single diagram, J.
Geophys. Res., 106(D7), 7183–7192, doi:10.1029/2000JD900719.
Thyng, K. M., Greene, C. A., Hetland, R. D., Zimmerle, H. M., & DiMarco, S. F. (2016). True
colors of oceanography. Oceanography, 29(3), 10.
Warner, J.C. et al. (2005). Performance of four turbulence closure models implemented using a
generic length scale method, Ocean Modelling, Volume 8, Issues 1–2, 2005, Pages 81-113.